# SAM transmethylation pathway and adenosine recycling to ATP are essential for systemic regulation and immune response

**Pavla Nedbalová[1], Nikola Kaislerova[1], Lenka Chodakova[1], Martin Moos[2,3]\*, Tomáš Doležal[1]\***

[1]Department of Molecular Biology and Genetics, Faculty of Science, University of South Bohemia, České Budějovice, Czech Republic; [2]Laboratory of Analytical Biochemistry and Metabolomics, Institute of Entomology, Biology Centre, Czech Academy of Sciences, České Budějovice, Czech Republic; [3]Department of Applied Chemistry, Faculty of Agriculture and Technology, University of South Bohemia, České Budějovice, Czech Republic

**\*For correspondence:**
moos@bclab.eu (MM);
tomas.dolezal@prf.jcu.cz (TD)

**Competing interest:** The authors declare that no competing interests exist.

## eLife Assessment

This paper provides a **valuable** contribution to our understanding of how adenosine acts as a signal of nutrient insufficiency and extends this idea to suggest that adenosine is released by metabolically active cells in proportion to the activity of methylation events. **Convincing** data supports this idea. The authors use metabolic tracing approaches to identify the biochemical pathways that contribute to the regulation of adenosine levels and the S-adenosylmethionine cycle in *Drosophila* larval hemocytes in response to wasp egg infection.

**Abstract** During parasitoid wasp infection, activated immune cells of *Drosophila melanogaster* larvae release adenosine to conserve nutrients for immune response. S-adenosylmethionine (SAM) is a methyl group donor for most methylations in the cell and is synthesized from methionine and ATP. After methylation, SAM is converted to S-adenosylhomocysteine, which is further metabolized to adenosine and homocysteine. Here, we show that the SAM transmethylation pathway is up-regulated during immune cell activation and that the adenosine produced by this pathway in immune cells acts as a systemic signal to delay *Drosophila* larval development and ensure sufficient nutrient supply to the immune system. We further show that the up-regulation of the SAM transmethylation pathway and the efficiency of the immune response also depend on the recycling of adenosine back to ATP by adenosine kinase and adenylate kinase. We therefore hypothesize that adenosine may act as a sensitive sensor of the balance between cell activity, represented by the sum of methylation events in the cell, and nutrient supply. If the supply of nutrients is insufficient for a given activity, adenosine may not be effectively recycled back into ATP and may be pushed out of the cell to serve as a signal to demand more nutrients.

## Introduction

Upon activation, immune cells fundamentally change their metabolism, requiring an increased supply of nutrients, which is essential for their effective function. Additional nutrients are delivered to

**eLife digest** When confronted with an infection, immune cells are rapidly activated to fight the threat. However, like all cells, they require energy to act. While most cells reduce their activity when nutrients are scarce, the immune system cannot afford to do so, as halting its response could put the entire body at risk from infection.

It is not clear how immune cells manage this complex nutritional budgeting. Previous studies of fruit fly larvae infected with a parasitoid wasp revealed that immune cells secure extra energy by releasing a molecule called adenosine. This slows the metabolism of non-immune tissues, leaving more nutrients available for immune cells. However, the exact mechanism that immune cells use to produce adenosine remained uncertain.

To further examine this process, Nedbalova et al. – who are part of the research group that carried out the previous work – extracted activated immune cells from a parasitoid-infected larva and fed them a labelled amino acid. Tracing this label revealed an increase in the number of chemical units known as methyl groups that had been added to molecules within the cell. This process, known as methylation, can regulate metabolic activity within cells and produces adenosine as a byproduct.

Further genetic studies showed that if nutrient supplies were sufficient, the immune cells recycled this adenosine back into ATP, the body's main energy currency. This suggests that if there were not enough nutrients to do this, the excess adenosine would slow the metabolism of non-immune cells, therefore securing more nutrients for the immune cells. Therefore, Nedbalova et al. hypothesise that these two processes could form the basis of a feedback mechanism that allows the immune cells to regulate their energy demands.

Taken together, the findings suggest that adenosine may act as a sensor to reflect immune activity, with it being released when the cells are stimulated and recycled if they have enough energy. This hypothesis still requires further testing but, as adenosine pathways are present across all organisms, it could have implications for many physiological and disease-related processes.

immune cells by redirecting metabolites away from other tissues, making the immune system metabolically privileged, or selfish within an organism. The concept of selfish immunity proposes insulin resistance as a mechanism by which activated immune cells ensure adequate nutrient supply (*Straub, 2014*; *Dolezal, 2015*; *Krejčová et al., 2023*). For example, we have recently shown that immune cells produce cytokines Upd2 and Upd3, which induce the production of the insulin signaling inhibitor Impl2 in muscle tissue via the Jak/Stat signaling pathway. Impl2 reduces the consumption of carbohydrates by muscle, making nutrients more available to immune cells (*McMullen et al., 2024*). Another mechanism by which immune cells gain privileged access to nutrients is through the release of adenosine. Extracellular adenosine signaling, through the adenosine receptor, reduces nutrient consumption by non-immune tissues and slows the development of the organism, again leaving more nutrients available for the immune system (*Bajgar et al., 2015*; *Dolezal, 2015*; *Bajgar and Dolezal, 2018*).

Adenosine signaling plays a variety of roles, especially in more complex organisms such as mammals. However, the release of adenosine as a signaling molecule from cells plays a universal role from the most primitive organisms, such as social bacteria, to humans (*Newby, 1984*; *Boison and Jarvis, 2021*). The release of adenosine is an indication of metabolic stress and signals to neighbouring cells to adjust their behaviour accordingly. For example, the bacterium *Myxococcus xanthus* responds to nutrient depletion by releasing adenosine, which initiates the aggregation of bacteria into fruiting bodies (*Shimkets and Dworkin, 1981*). In mammals, the heart muscles secrete adenosine during ischemia, which causes blood vessels to dilate to increase blood flow and delivery of oxygen and nutrients (*Lloyd et al., 1988*; *Deussen et al., 1989*; *Kroll et al., 1992*; *Reiss et al., 2019*). Adenosine also plays an important role in the induction of sleep, torpor and hibernation, physiological states associated with energy conservation (*Fredholm et al., 2011*; *Boison and Jarvis, 2021*). Therefore, the use of extracellular adenosine by immune cells to suppress nutrient consumption in non-immune tissues, thereby conserving energy for the immune response, fits well with this evolutionarily conserved role of adenosine signaling.

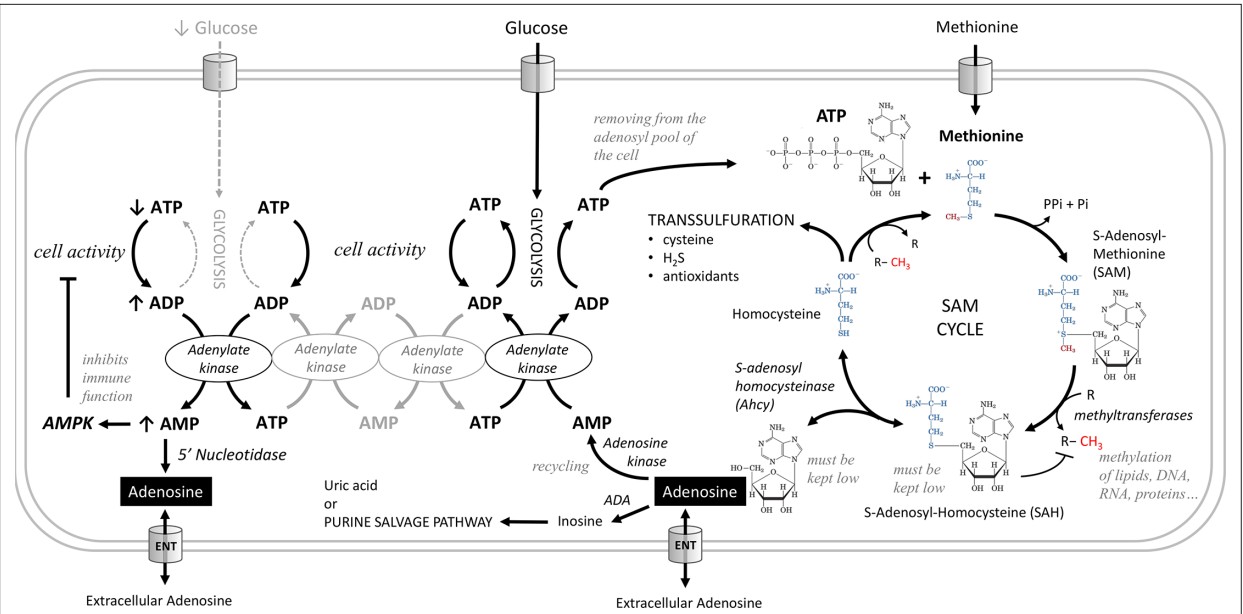

**Figure 1.** Metabolic pathways producing adenosine. Adenosine can be produced from AMP by 5'-nucleotidase. AMP increases during metabolic stress, when cells use adenylate kinase to make one ATP and one AMP from two ADP molecules. Increased AMP activates AMP-activated protein kinase (AMPK), which suppresses energy-consuming processes in the cell and activates energy-producing processes, or is converted to adenosine, which can be released from cells via the equilibrative nucleoside transporter (ENT). In the S-adenosylmethionine (SAM) transmethylation pathway, the combination of ATP and methionine produces SAM, the major methyl group donor (red) for the majority of methylations that occur in the cell. During methylation, SAM is converted to S-adenosylhomocysteine (SAH), which is rapidly converted to homocysteine and adenosine by adenosylhomocysteinase (Ahcy), as it would otherwise block further methylation. Ahcy works bidirectionally, and for the 'SAM to homocysteine +adenosine' direction to dominate, homocysteine and adenosine must be rapidly metabolized or cleared from the cell. Homocysteine is remethylated back to methionine or further metabolized via the transsulfuration pathway (source of cysteine, $H_2S$, and antioxidants). The catabolism of adenosine to inosine is mediated by adenosine deaminase (ADA) or recycled to AMP by adenosine kinase which becomes ATP through adenylate kinase, glycolysis or oxidative phosphorylation. Another possible source of adenosine is RNA degradation, which is not shown in this scheme.

Adenosine is produced intracellularly in two ways (*Figure 1*), by conversion from AMP by nucleotidases or in the S-adenosylmethionine (SAM) transmethylation pathway from S-adenosylhomocysteine (SAH) by S-adenosylhomocysteinase (Ahcy). The increase in AMP mainly reflects the lack of energy and metabolic stress. Cells use adenylate kinase to produce one ATP and one AMP from two ADP molecules. Increased AMP activates AMPK, which suppresses energy-consuming processes in the cell and activates energy-producing processes (*Kloor and Osswald, 2004*; *Dzeja and Terzic, 2009*). AMP is also converted by nucleotidases to adenosine, which is released from the cell via the equilibrative nucleoside transporter (ENT), where it signals the metabolic stress of the cell to surrounding cells and tissues via adenosine receptors (*Dolezelova et al., 2007*; *Fenckova et al., 2011*). This is typical of cells during ischemia when they are deprived of nutrients (*Deussen et al., 1989*; *Schädlich et al., 2023*). Since activated AMPK suppresses the function of immune cells (*Salminen et al., 2011*), we would expect that in the absence of nutrients and decreasing ATP levels, the accumulated AMP would be converted to adenosine and released from the cells, thus demanding a greater supply of nutrients.

In the SAM transmethylation pathway, ATP and methionine combine to produce SAM, which is the major methyl group donor for most methylations in the cell. The SAM transmethylation pathway is one of the largest consumers of ATP when the adenosyl moiety of ATP is incorporated into SAM (i.e. it is not a conversion of ATP to ADP as in energy transfer). After the methyl group is transferred to the target molecule by methyltransferase, SAM becomes SAH, which must be rapidly converted to homocysteine and adenosine, or it will block further methylation. As methylation increases in the cell, more SAM is formed and at the same time SAH is cleared more rapidly, leading to an increase in the SAM:SAH ratio, known as the methylation index. SAH is converted to homocysteine and adenosine by Ahcy, which works in both directions; for the 'SAM to homocysteine +adenosine' direction to prevail, homocysteine and adenosine must also be metabolized or excreted from the cell very rapidly (*Lu, 2000*; *Kloor and Osswald, 2004*; *Yu et al., 2019*; *Roy et al., 2020*; *Vizán et al., 2021*).

Homocysteine can either be recycled back to methionine, or further metabolized in the transsulfuration pathway (source of cysteine, $H_2S$, and antioxidants) (*Sbodio et al., 2019*). Adenosine can be catabolized to inosine by adenosine deaminase (*Novakova and Dolezal, 2011*), or recycled to AMP by adenosine kinase and further to ATP by adenylate kinase, glycolysis or oxidative phosphorylation (*Zeleznikar et al., 1990*; *Zeleznikar et al., 1995*; *Dzeja et al., 2007*; *Dzeja and Terzic, 2009*). The importance of adenosine recycling to AMP by adenosine kinase is indicated by the dependence of the methylation cycle on the concomitant function of this enzyme (*Lecoq et al., 2001*; *Bjursell et al., 2011*; *Moffatt et al., 2002*). Since every single methylation in the cell is associated with the formation of one adenosine molecule, the amount of adenosine accurately reflects the intensity of the SAM transmethylation pathway. In a sense, it reflects the activity of the cell, because the more processes the cell has to activate, the more new molecules it has to methylate.

We have previously shown that activated immune cells preferentially acquire nutrients during immune response by releasing adenosine, subsequently reducing nutrient uptake in non-immune tissues, thereby leading to a delay in development (*Bajgar et al., 2015*; *Bajgar and Dolezal, 2018*). To elucidate the origin of adenosine in activated immune cells, we investigated the role of the SAM transmethylation pathway. Given its upregulation and critical function in mammalian immune cells (*Lawson et al., 2012*; *Yu et al., 2019*; *Roy et al., 2020*), we hypothesized that this pathway is also a major source of adenosine in *Drosophila* immune cells. Our results confirm that the SAM transmethylation pathway is indeed upregulated in *Drosophila* hemocytes upon parasitoid wasp infection. In addition, we show that this pathway is not only essential for the immune response, but also a source of extracellular adenosine that slows larval development. Moreover, not all adenosine is released from hemocytes, but it can be recycled back into ATP and thus re-enter the SAM transmethylation pathway.

## Results

### SAM transmethylation pathway accelerates in activated hemocytes

One of the important sources of adenosine in hemocytes is the SAM transmethylation pathway (*Figure 2A*). Putative methionine transporters, as well as all enzymes of this pathway, are strongly expressed in hemocytes, both in the resting and activated state (*Figure 2B*, *Figure 2—figure supplement 1*, *Figure 2—figure supplement 1—source data 1* and *Supplementary file 3*). Methionine can be transported from the hemolymph to the cytosol by *Drosophila* L type amino acid transporters orthologs mnd (CG3297; FBgn0002778), Jhl-21 (CG12317; FBgn0028425), CG1607 (FBgn0039844), sbm (CG9413; FBgn0030574), or gb (CG6070; FBgn0039487). The expression of *mnd* and *gb* is particularly high and is further increased during infection (*Figure 2B*). SAM synthetase (Sam-S; CG2674; EC 2.5.1.6; FBgn0005278) converts methionine and ATP into SAM. SAM is used for methylation by a wide range of methyltransferases (EC 2.1.1.-), or in polyamine synthesis, by SAM decarboxylase (SamDC; CG5029; EC 4.1.1.50; FBgn0019932), producing 5'-methylthioadenosine (MTA). The actual methylation, using SAM as a donor, is carried by various methyltransferases. Using our RNAseq of hemocytes, we checked the expression of 200 genes (S3 Table), which are categorized as *Drosophila* methyltransferases. We found that 43 methyltransferases (at least 24 of them categorized as SAM-dependent methyltransferases) significantly increased their expression in both 9 and 18 hpi (*Figure 2—figure supplement 3*). Methylation reactions produce SAH, a potent inhibitor of methyltransferases which must be quickly removed to keep the pathway running. Ahcy (Ahcy; CG11654; EC 3.3.1.1; FBgn0014455) hydrolyzes SAH into adenosine and homocysteine and shows the highest expression among SAM transmethylation pathway enzymes (*Figure 2B*). AhcyL1 (CG9977; FBgn0035371) and AhcyL2 (CG8956; FBgn0015011), which work as dominant-negative regulators of Ahcy *Parkhitko et al., 2016* have a lower expression and are downregulated during infection (*Figure 2B*). When adenosine and homocysteine accumulate, Ahcy changes the activity in favor of SAH synthesis. Thus, adenosine and homocysteine must be rapidly metabolized (*Kloor and Osswald, 2004*). Homocysteine can be remethylated back to methionine, by betaine-homocysteine methyltransferase (Bhmt; CG10623; EC 2.1.1.5; FBgn0032727) or homocysteine S-methyltransferase (CG10621; EC 2.1.1.10; FBgn0032726). Alternatively, it is used as a substrate for Cystathionine β-synthase (Cbs; CG1753; EC 4.2.1.22; FBgn0031148) to produce cystathionine (CTH) in the transsulfuration pathway (*Figure 2A*). Overall, enzymes of the SAM transmethylation pathway, and its associated metabolic branches, are strongly expressed in hemocytes (*Figure 2B*) and possible increase in SAM transmethylation pathway

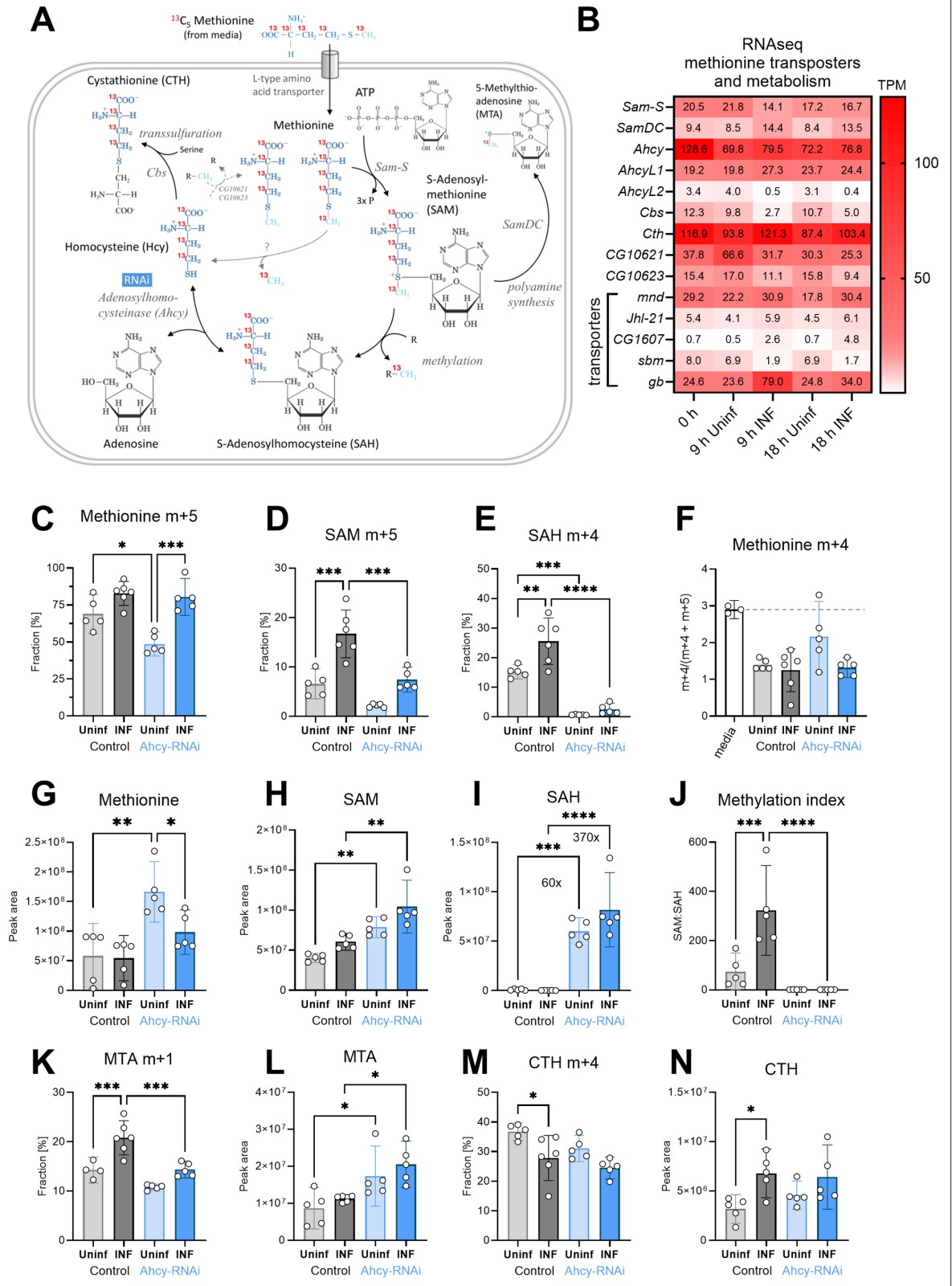

**Figure 2.** Analysis of the SAM transmethylation pathway in larval hemocytes by ex vivo stable $^{13}$C isotope tracing. (**A**) Schematic representation of the SAM transmethylation pathway with polyamine synthesis and transsulfuration branches and labeling with L-methionine-$^{13}$C$_5$ from media (red 13 represents the labeled carbon). Light blue CH$_3$ represents the methyl group used during transmethylation or remethylation. Enzymes and processes are italicized. RNAi in blue rectangle represents adenosylhomocysteinase knockdown. (**B**) Expression heat map (bulk RNAseq) of methionine transporters

*Figure 2 continued on next page*

*Figure 2 continued*

and enzymes in circulating hemocytes from uninfected (Uninf) and infected (INF) third instar larvae collected at 0, 9 and 18 hpi (0 hpi = 72 hr after egg laying). Means of 6 replicates (10 replicates in case of 18 hr INF) shown in each cell are transcripts per million (TPM) - for comparison, only 20% of genes in RNAseq show expression higher than 15 TPM (data in *Figure 2—source data 1* and *Supplementary file 3*). (C-E,K,M) $^{13}$C-labeling of metabolites in hemocytes, which were incubated ex vivo for 20 min in media containing 0.33 µM L-methionine-$^{13}$C$_5$. The graphs show the fraction of the compound with one (m+1), four (m+4), or five (m+5) $^{13}$C-labeled carbons – methionine m+5 (**C**), S-adenosylmethionine (SAM) m+5 (**D**), S-adenosylhomocysteine (SAH) m+4 (**E**), 5-methylthioadenosine (MTA) m+1 (**K**) and cystathionine (CTH) m+4 (**M**). (**F**) Methionine m+4 portion of total labeled methionine (m+4 and m+5) in media representing the labeling impurity of used methionine (white bar setting threshold - dashed line), and in hemocyte samples. (**G-I,L,N**) Total levels of methionine (**G**), SAM (**H**), SAH (**I**), MTA (**L**), and CTH (**N**) in hemocytes shown as the mean metabolite amounts expressed by the normalized peak area. (**J**) Methylation index calculated as the ratio of SAM:SAH levels (peak areas in H and I). (**C-N**) Bars represent mean values with 95% CI of uninfected (Uninf, light grey) and infected (INF, dark grey) control and uninfected (Uninf, light blue) and infected (INF, dark blue) Ahcy-RNAi samples; each dot represents one biological replicate (numerical values in *Figure 2—source data 1* and *Supplementary file 1*); asterisks represent significant differences between samples tested by ordinary one-way ANOVA Tukey's multiple comparison test (*$p<0.05$, **$p<0.01$, ***$p<0.001$, ****$p<0.0001$).

The online version of this article includes the following source data and figure supplement(s) for figure 2:

**Source data 1.** The MS Excel file containing the raw data/numerical values used to generate the plots in the figure.

**Figure supplement 1.** Expression of SAM transmethylation pathway enzymes and Ahcy RNAi.

**Figure supplement 1—source data 1.** The MS Excel file containing the raw data/numerical values used to generate the plots in the figure.

**Figure supplement 2.** Srp-Gal4 Gal80 driver expression.

**Figure supplement 3.** Expression of selected methyltransferases.

**Figure supplement 3—source data 1.** The MS Excel file containing the raw data/numerical values used to generate the plots in the figure.

can be suggested from RNAseq data (1) by increased expression of four transporters capable of transporting methionine, (2) by decreased expression of AhcyL2 (dominant-negative regulator of Ahcy), and (3) by increased expression of 43 out of 200 methyltransferases. Nevertheless, the actual metabolism cannot be determined based solely on the gene expression.

To analyze SAM transmethylation pathway metabolism, we used high pressure liquid chromatography coupled with tandem mass spectrometry (HPLC-MS/MS). In addition to detecting relative levels of metabolites, 13C-labeled methionine tracing was used to determine methionine usage in the SAM transmethylation pathway in hemocytes of infection and non-infected animals. Hemocytes were extracted by bleeding larvae at 20 hours post infection (hpi). Cells were incubated in media containing 0.33 µM $^{13}$C$_5$ methionine for 20 min. This ex vivo incubation was designed to resemble physiological conditions. In control larvae, the labeled intracellular methionine fraction reached almost 70% in resting hemocytes and over 80% in activated hemocytes after 20 min of ex vivo incubation, demonstrating a substantial uptake of methionine into hemocytes (*Figure 2C*). Much of this methionine is used for SAM production and methylation, as indicated by the twofold increase of SAM and SAH in infected compared to control animals (*Figure 2D and E*); demonstrating an upregulation of the SAM transmethylation pathway in activated hemocytes. This is further supported by a significantly higher methylation index (ratio of SAM:SAH; *Figure 2H–J*) in activated hemocytes.

SAH is cleaved into homocysteine and adenosine; however, we were unable to reliably detect homocysteine in our metabolomic analysis. Homocysteine can be recycled back to methionine, to enter another round of the SAM transmethylation pathway. In this scenario, fully labeled 13C5 methionine, as was used in this assay, would lead to the formation of methionine m+4. One labeled carbon is used in the methylation reaction while the unlabeled methyl group would be added during recycling (*Figure 2A*). However, the m+4 portion of the labeled methionine (m+4 and m+5) did not exceed the 3% threshold, set by the methionine labeling impurity (ratio 'm+4/[m+4 + m+5]' in the medium was 3%; *Figure 2F*). This indicates that methionine used for the SAM transmethylation pathway in hemocytes is derived from hemolymph and homocysteine is not remethylated to methionine.

To investigate the role of the SAM transmethylation pathway in hemocytes, we knocked down Ahcy by hemocyte-specific RNAi (*Figure 2—figure supplements 1 and 2*). Ahcy RNAi leads to reduced 13C-labeled methionine uptake in resting hemocytes, although uptake is still increased upon infection (*Figure 2C*). Methionine incorporation into SAM is significantly lower in activated hemocytes (*Figure 2D*) and incorporation into SAH is decreased in both, quiescent and activated hemocytes (*Figure 2E*). These results imply that the SAM transmethylation pathway slows down when there is insufficient Ahcy available. This is reflected in metabolite levels, where SAH is heavily accumulated in

Ahcy RNAi hemocytes (*Figure 2I*) and SAM and methionine levels are higher compared to controls (*Figure 2G and H*); indicating that they are not metabolized in the SAM transmethylation pathway. This is substantiated by a methylation index that is very low compared to controls, and does not increase upon infection (*Figure 2J*).

SAM can also be utilized for polyamine synthesis that, in addition to polyamines, produces 5-methylthioadenosine (MTA). Polyamines are essential for cell growth and proliferation and participate in various cellular processes (replication, gene expression, biomolecules modifications, etc.). They play a role in macrophage polarization and regulation (*Latour et al., 2020*; *Xuan et al., 2023*). There is significantly higher incorporation of 13 C methionine into MTA in activated hemocytes (*Figure 2K*) with no changes in MTA level (*Figure 2L*). Ahcy RNAi hemocytes have increased MTA levels (*Figure 2L*), which can reflect MTA accumulation, due to decreased SAM transmethylation pathway activity. Decreased 13 C methionine incorporation into MTA upon Ahcy knockdown by RNAi (*Figure 2K*) may be caused by dilution, as unlabeled MTA accumulate (*Figure 2L*) therefore, we are unable to determine the effect of Ahcy RNAi on polyamine synthesis.

The SAM transmethylation pathway is connected to the transsulfuration pathway, which produces cysteine and further antioxidants taurine, and glutathione (*Sbodio et al., 2019*). The transsulfuration pathway is one of the ways to metabolically remove homocysteine produced by Ahcy when Cbs converts homocysteine into CTH. This is supported by notable 13 C methionine incorporation into CTH (*Figure 2M*). Interestingly, there is a small, but significant, drop in the labeled CTH fraction in activated hemocytes (*Figure 2M*), however, this could be due to an overall increase in the level of CTH, including the unlabeled form (*Figure 2N*). As reported above, the SAM transmethylation pathway is accelerated in activated hemocytes, producing more adenosine, as well as more homocysteine, which must be removed to keep the pathway running. Cystathionine γ-lyase (Cth; CG5345; EC 4.4.1.1; FBgn0000566) usually converts CTH to cysteine. However, this enzyme can also use homocysteine for $H_2S$ production, and was shown to be responsible for homocysteine removal under hyperhomocysteinemia, when Cbs is unresponsive (*Singh et al., 2009*). $H_2S$ has many diverse roles in the organismal physiology, dependent on its source, localization, and concentration, and has already been connected with immune cells (*Sbodio et al., 2019*; *Dilek et al., 2020*). Based on our transcriptomic data, *Cbs* expression in *Drosophila* larval hemocytes is rather low, and even decreases during infection. Interestingly, cystathionine γ-lyase is strongly expressed, at similar levels to Ahcy. Therefore, our data gives evidence to the clearance of homocysteine by cystathionine γ-lyase and $H_2S$ production rather than via CTH and Cbs.

Knockdown of Ahcy leads to the accumulation of SAH and thus the production of homocysteine, the substrate for CTH, should be strongly inhibited. However, our data shows the relative levels of methionine, as well as the incorporation of 13 C methionine into CTH in Ahcy RNAi hemocytes is equivalent to controls (*Figure 2M and N*). This suggests, there is an alternative connection of methionine with homocysteine and consequently CTH, in addition to Ahcy. Uncovering this alternative connection would be of interest, although it is beyond the scope of this current work.

In summary, activated hemocytes increase the uptake of methionine to ramp up the transmethylation pathway. SAM is not only a substrate for methylation, but is also used for polyamine synthesis. Homocysteine is not recycled back to methionine, but is further metabolized in the transsulfuration pathway. Interestingly, our data suggest an alternative link between methionine and CTH independent of Ahcy. Hemocyte-specific Ahcy-RNAi potently suppresses the SAM transmethylation pathway.

## Ahcy produces adenosine as a systemic signal that influences the immune response

In our previous work, we showed that activated hemocytes release increased amount of adenosine (*Bajgar et al., 2015*; *Bajgar and Dolezal, 2018*). This causes a systemic metabolic switch that slows the organism's development, leaving energy for an effective immune response. Our results show that the SAM transmethylation pathway is enhanced in hemocytes after infection. This means that increased levels of adenosine is produced, as every methylation event results in the production of one adenosine molecule. At the same time, the cell must rapidly remove adenosine, along with homocysteine, to avoid reversing the direction of Ahcy enzyme activity, which would block methylation. This appears to be reflected in the significant reduction in intracellular adenosine levels after infection (*Figure 3A*). At least part of this reduction is likely due to the release of adenosine from the cell (via

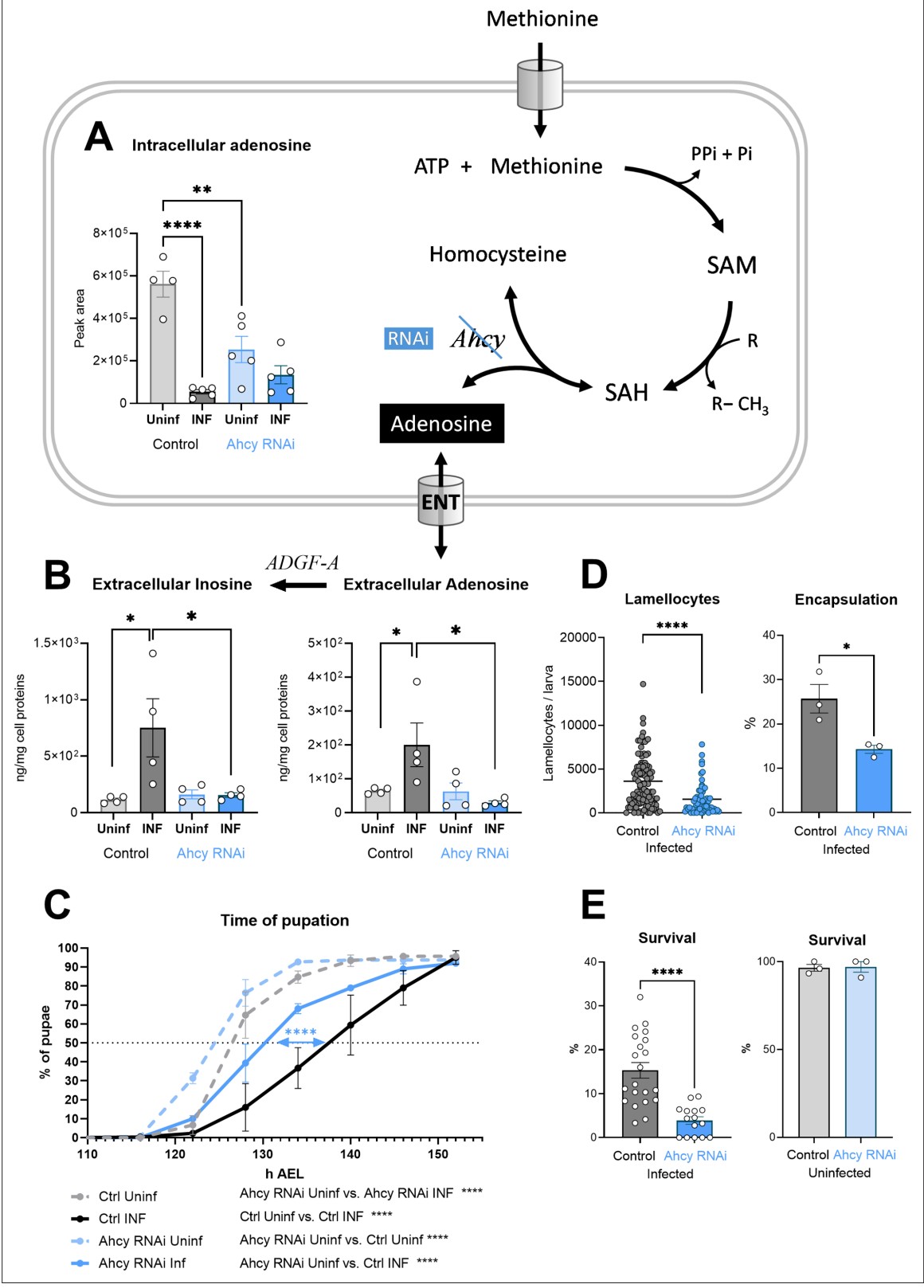

**Figure 3.** Generation of adenosine in the SAM transmethylation pathway and its systemic effects. (**A**) Levels of intracellular adenosine in hemocytes shown as the mean metabolite amount expressed by the normalized peak area at 20 hpi. Infection significantly decreases the level in control hemocytes (uninfected - Uninf - light gray vs. infected - INF - dark gray). Hemocyte-specific adenosylhomocysteinase knockdown (Ahcy RNAi; uninfected - Uninf - light blue and infected - INF - dark blue) significantly decreases intracellular adenosine in the uninfected state. (**B**) Levels of extracellular adenosine

*Figure 3 continued on next page*

*Figure 3 continued*

(right), released from hemocytes ex vivo after 20 min, and extracellular inosine (left) generated by adenosine deaminase ADGF-A. While infection leads to increased release of adenosine and generated inosine in the control (gray), no such increase is detected with Ahcy RNAi (blue). (**C**) Pupation is delayed upon infection in control larvae (n=270, uninfected dashed gray line and n=240, infected solid black line) but significantly less in hemocyte-specific Ahcy RNAi larvae (n=260, uninfected dashed blue line and n=265, infected solid blue line). Lines represent percentages of pupae at hours after egg laying (h AEL); rates were compared using Log-rank survival analysis. (**D**) The number of lamellocytes as well as encapsulation efficiency is significantly lower in infected Ahcy-RNAi (blue) larvae compared to infected control (gray). (**E**) Percentage of infected larvae surviving to adulthood is significantly lower in Ahcy-RNAi (blue) compared to control (gray) while the survival of uninfected individuals is not affected. (**A,B,E**) Bars represent means with SEM of uninfected (Uninf, light gray) and infected (INF, dark gray) control and uninfected (Uninf, light blue) and infected (INF, dark blue) Ahcy-RNAi samples; each dot represents a biological replicate; (**D**) dots represent the number of lamellocytes in a larva and the line mean; numerical values are in *Figure 3—source data 1* and *Supplementary file 1*; asterisks represent significant differences between samples tested and unpaired t test (*p<0.05, **p<0.01, ***p<0.001, ****p<0.0001).

The online version of this article includes the following source data for figure 3:

**Source data 1.** The MS Excel file containing the raw data/numerical values used to generate the plots in the figure.

the equilibrative nucleoside transporter ENT2 *Bajgar et al., 2015*) since extracellular adenosine levels increase after infection (*Figure 3B*), as does the level of inosine, which is formed from adenosine by the action of the adenosine deaminase ADGF-A. We measured the extracellular adenosine production in hemocytes after 20 min of ex vivo incubation, when hemocytes were removed by centrifugation and adenosine and inosine were measured in the collected supernatants. We also chose to detect inosine because activated hemocytes produce the adenosine deaminase ADGF-A (*Novakova and Dolezal, 2011*). Therefore, some adenosine is always converted to inosine during sample processing, even when the adenosine deaminase inhibitor EHNA (erythro-9-(2-hydroxy-3-nonly)adenine) is used, otherwise it is difficult to detect any adenosine.

Hemocyte-specific Ahcy RNAi leads to a significant decrease in intracellular adenosine levels, confirming that a significant fraction of adenosine is generated by the SAM transmethylation pathway (*Figure 3A*). However, there is no increase in extracellular adenosine, or inosine levels upon Ahcy knockdown in hemocytes following infection (*Figure 3B*). This indicates that the SAM transmethylation pathway is responsible for the generation of this extracellular signal. This is further supported by the fact that silencing of Ahcy in hemocytes during infection leads to a significant reduction (by 6 hr) in developmental delay (*Figure 3C*), the hallmark of systemic effects of adenosine during infection. Similar to our previous work (*Bajgar et al., 2015*), this results in diminished host defense, which is associated with reduced production of lamellocytes and encapsulation of the parasitoid (*Figure 3D*) and reduced survival (*Figure 3E*). Interestingly, the induction of Ahcy RNAi in hemocytes accelerates development by approximately 2 hr, even in the absence of infection (*Figure 3C*), a result similar to silencing ENT2 specifically in hemocytes (*Bajgar et al., 2015*). The normal activity of hemocytes, associated with methylation and adenosine release, appears to slow the overall development of the organism.

In summary, the SAM transmethylation pathway and Ahcy produce the majority of adenosine in *Drosophila* larval hemocytes and strongly participate in the production of extracellular adenosine as a systemic metabolic regulator. Knockdown of Ahcy in hemocytes reduces larval developmental delay associated with infection, demonstrating the lack in extracellular adenosine production. In addition, knockdown of Ahcy in hemocytes negatively impacts immune response.

## Activated hemocytes recycle ATP and SAM from adenosine

The above results demonstrate that the SAM transmethylation pathway is a major producer of adenosine in activated hemocytes. To maintain methylation, the cell must rapidly remove adenosine, some of which is released from the cell. This, of course, reduces the adenosyl pool of the cell; note that the SAM pathway is a major consumer of ATP when it combines methionine with ATP to form SAM (*Figure 4A*). Therefore, activated immune cells have active de novo purine production, as shown in mammalian macrophages and T lymphocytes (*Yu et al., 2019*; *Roy et al., 2020*). Our recent work showed increased pentose phosphate pathway activity is linked with nucleotide production in activated hemocytes (*Kazek et al., 2024*). However, this is very energy consuming and requires many steps, whereas the recycling of adenosine to AMP requires only one step and one ATP. Ahcy activity has indeed been linked to adenosine kinase activity, which produces AMP from Ado (*Moffatt et al.,*

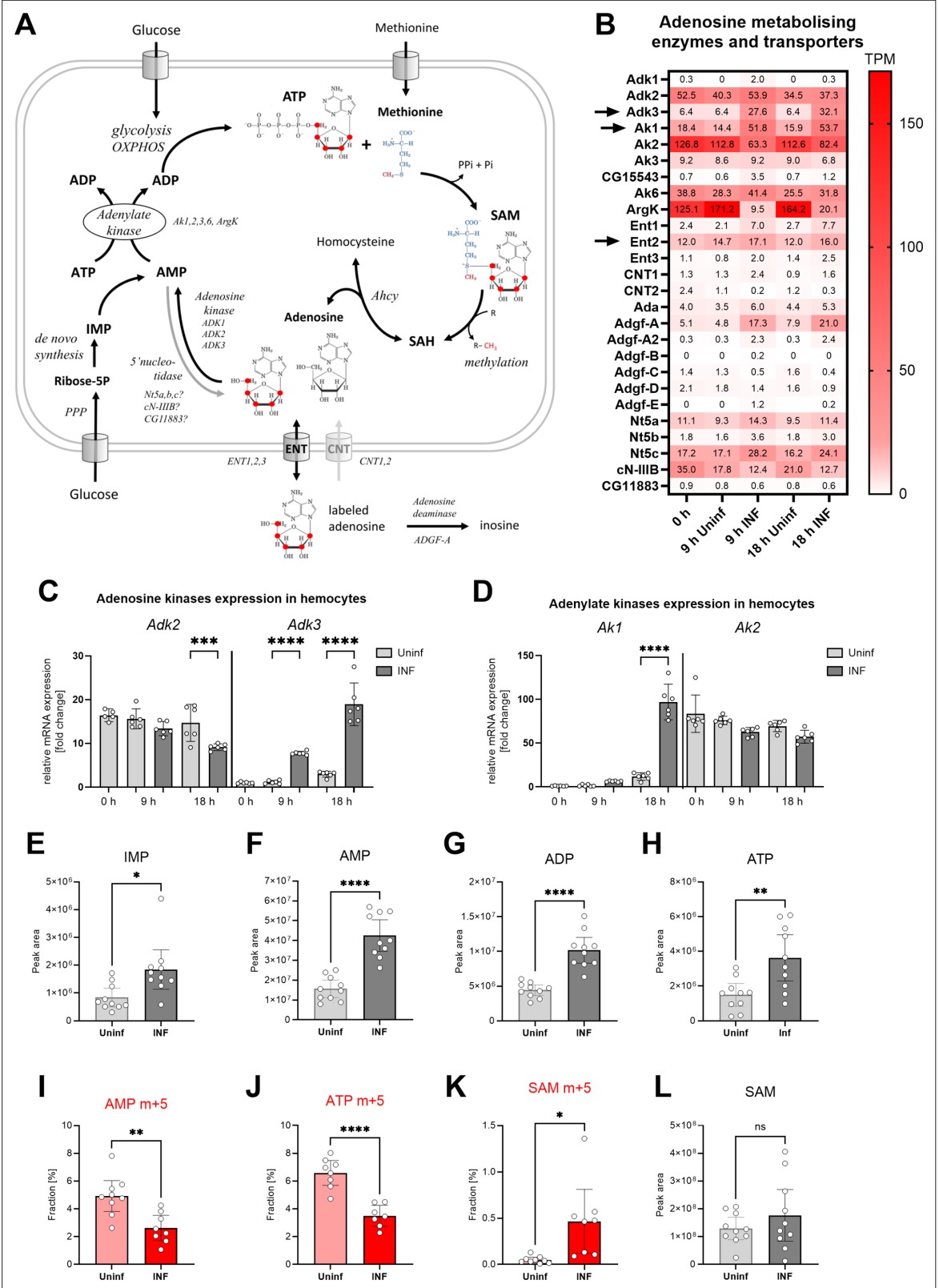

**Figure 4.** Analysis of adenosine recycling to SAM in larval hemocytes by ex vivo stable $^{13}$C isotope tracing. (**A**) Schematic representation of the SAM transmethylation pathway, de novo purine synthesis and adenosine recycling to ATP and SAM and labeling with adenosine-$^{13}$C$_5$ from media (red 13 represents the labeled carbon). Enzymes and processes are italicized. (**B**) Expression heat map (bulk RNAseq) of adenosine metabolizing enzymes and transporters in circulating hemocytes from uninfected (Uninf) and infected (INF) third instar larvae collected at 0, 9, and 18 hpi (0 hpi = 72 hr after egg

*Figure 4 continued on next page*

Figure 4 continued

laying). Means of 6 replicates (10 replicates in case of 18 hr INF) shown in each cell are transcripts per million (TPM) - for comparison, only 20% of genes in RNAseq show expression higher than 15 TPM (data in *Figure 4—source data 1* and *Supplementary file 3*). (C, D) Expression analysis of adenosine kinases Adk2, 3 (C) and adenylate kinases Ak1, 2 (D) in circulating hemocytes from uninfected (Uninf) and infected (INF) third instar larvae collected at 0, 9, and 18 hpi by RT-qPCR. Bars show fold change compared to 0 hr Adk3 samples (expression levels were normalized to *RpL32* expression in each sample), each dot represents a biological replicate. (E-H, L) Total levels of IMP (E), AMP (F), ADP (G), ATP (H) and S-adenosylmethionine - SAM (L) in hemocytes shown as the mean metabolite amounts expressed by the normalized peak area. (I-K) $^{13}$C-labeling of metabolites in hemocytes, which were incubated ex vivo for 20 min in media containing 10 μM adenosine-$^{13}$C$_5$. The graphs show the fraction of the compound with five $^{13}$C-labeled carbons – AMP m+5 (I), ATP m+5 (J) and SAM m+5 (K). (C-L) Bars represent mean values with 95% CI of uninfected (Uninf, light gray or pink) and infected (INF, dark gray or red) samples; each dot represents a biological replicate (numerical values in *Figure 4—source data 1* and *Supplementary file 1*); asterisks represent significant differences between samples tested by tested by ordinary one-way ANOVA Sidak's multiple comparison test (**C,D**) and unpaired t test (**E-L**) (*p<0.05, **p<0.01, ***p<0.001, ****p<0.0001).

The online version of this article includes the following source data for figure 4:

**Source data 1.** The MS Excel file containing the raw data/numerical values used to generate the plots in the figure.

*2002*; *Xu et al., 2017*; *Murugan et al., 2021*). AMP can then be converted to ADP by adenylate kinase (*Dzeja and Terzic, 2009*) and finally to ATP by glycolysis or OXPHOS, which can again enter the transmethylation pathway.

The most striking expression change we detected in activated hemocytes was the increase of adenosine kinase Adk3 (*Figure 4B and C*) and cytosolic adenylate kinase Ak1 (*Figure 4B and D*). Another adenosine kinase, Adk2, which is not altered by infection, is also highly expressed in hemocytes, as is the mitochondrial adenylate kinase Ak2 (*Figure 4B–D*). Activated hemocytes have a system for recycling adenosine back to ATP, which appears to be further enhanced during infection. The increase of cytosolic Ak1 indicates at least partial redirection of energy metabolism from mitochondria (oxidative phosphorylation) into cytosol (aerobic glycolysis) in activated immune cells (*Kornberg, 2020*).

The increase in all levels of IMP, AMP, ADP, and ATP (*Figure 4E–H*) indicates that de novo nucleotide production occurs in activated hemocytes. De novo purine synthesis may play many different roles in the cell, but an important one is the replenishment of ATP for SAM synthesis. In addition to de novo synthesis, our expression analysis suggests that recycling of adenosine to ATP is also involved. We tested this by adding labeled adenosine to resting and activated hemocytes ex vivo (*Figure 4A*). Since hemocytes express ENT2 (*Figure 4B* and *Bajgar et al., 2015*), at least some of the labeled adenosine enters the hemocytes and is converted to AMP and finally to ATP (*Figure 4I and J*). This demonstrates that hemocytes recycle adenosine to ATP, in addition to de novo purine synthesis. Both the AMP m+5 and ATP m+5 fractions are lower after infection (*Figure 4I and J*). This is likely due to dilution of the labeled fraction by a significant increase in newly synthesized, unlabeled AMP and ATP (*Figure 4F and H*). Activated hemocytes also release more adenosine from the cell, so it is possible that it is more difficult for labeled adenosine to enter the cell. Therefore, the extent of ATP recycling from adenosine in resting and activated hemocytes remains elusive.

Resting hemocytes show a diminished fraction of labeled SAM derived from labeled adenosine (*Figure 4K*). After infection, there is a significant increase in SAM labeling (*Figure 4K*), with approximately 13% of labeled ATP molecules being used for SAM synthesis (compare *Figure 4J* to *Figure 4K*), while the total level of SAM does not change (*Figure 4L*). Since ATP is labeled to greater extent in resting hemocytes, this suggests a compartmentalization of the enzymatic cascade physically linking SAM to adenosine recycling (*Pedley and Benkovic, 2017*; *Bar-Peled and Kory, 2022*). If any ATP was used for SAM production then a higher fraction of labeled ATP in resting hemocytes should be reflected in higher labeled SAM fractions. A higher fraction of labeled SAM in activated hemocytes suggests a physical connection of SAM synthesis and ATP recycling from labeled adenosine. Increased recycling of adenosine to ATP would be characterized by more 13 C entering SAM, despite the fraction of labeled ATP appearing lower due to the total amount of ATP produced by de novo synthesis.

In conclusion, both resting and activated hemocytes recycle adenosine to ATP. Activated hemocytes strongly upregulate the expression of Adk3 and cytosolic Ak1 and increase production of SAM from recycled adenosine.

## Adenosine kinase and cytoplasmic adenylate kinase are important for SAM pathway and immune response

To determine the importance of adenosine kinase and adenylate kinase, which seem to be necessary for recycling of ATP from adenosine, we again used hemocyte-specific RNAi induced by Srp-Gal4Gal80. Adk3 RNAi reduces the infection-induced increase in expression almost eight times (*Figure 5—figure supplement 1*). Adk3 expression is low in hemocytes of uninfected larvae (*Figure 4B and C*), thus we did not observe any effect of Adk3 RNAi on the AMP level in uninfected larvae (*Figure 5A*). However, there is significantly lower level of AMP in hemocytes from the infected larvae (*Figure 5A*). Adk2 is also strongly expressed both in resting and activated hemocytes (*Figure 4C*) and that is, most likely, why we still see an infection-induced increase in AMP upon Adk3 RNAi (*Figure 5A*). This increase is also due to de novo purine synthesis. Nevertheless, the increase in AMP in Adk3 RNAi activated hemocytes is significantly smaller than that of controls, indicating that Adk3 activity is important for adenosine conversion to AMP in activated hemocytes. Increase in methylation index is abrogated upon Adk3 RNAi (*Figure 5B*), showing that Adk3 activity is required for the infection-induced increase in SAM transmethylation pathway. The consequences of Adk3 knockdown phenocopy those of Ahcy knockdown animals, as development is significantly less delayed during infection (*Figure 5C*) and number of lamellocytes (*Figure 5D*), as well as survival (*Figure 5E*) are decreased. We observe similar effects upon cytoplasmic Ak1 knockdown in hemocytes (*Figure 5—figure supplement 1*), another enzyme upregulated upon infection, which is likely involved in adenosine recycling. The methylation index is abrogated (*Figure 5B*), development is significantly less delayed during infection (*Figure 5F*), number of lamellocytes is decreased (*Figure 5G*) and survival is lower (*Figure 5H*) upon Ak1 RNAi-mediated knockdown. In summary, recycling of adenosine to AMP and potentially to ATP and SAM, using adenosine kinase and cytoplasmic adenylate kinase, appears to be important for activated hemocytes to increase the SAM transmethylation pathway and for effective immune response.

## Discussion

We have previously shown that adenosine, released from *Drosophila* hemocytes, is required for systemic metabolic switch and to slow development so that there are enough nutrients for an effective immune response (*Bajgar et al., 2015*). Here, we show that this signal is generated in hemocytes by the SAM transmethylation pathway, which is enhanced upon infection and is required for the immune response. This novel observation links SAM transmethylation in immune cells to the production of adenosine as a systemic metabolic regulator. In addition, we show that adenosine is not only released from hemocytes but is also recycled to ATP and SAM, mediated by Adk3 and the Ak1, which are necessary for the increase in SAM transmethylation during infection, and therefore effective immune response.

The SAM transmethylation pathway is part of the one-carbon metabolism interconnecting the metabolism of serine, glycine, folates, and methionine, supporting de novo synthesis of purines and thymidine, and maintaining redox balance. The SAM transmethylation pathway itself controls antioxidant synthesis, sulfur metabolism, polyamine synthesis, and epigenetic programming via methylation (*Ducker and Rabinowitz, 2017*; *Vizán et al., 2021*). Therefore, Ahcy activity, removing SAH and producing adenosine and homocysteine, is essential, not only for keeping proper function of diverse methyltransferases, but also for a wide range of other essential metabolic processes inside the cell.

There is increasing evidence showing the crucial role of the SAM transmethylation pathway in immune cells (*Lawson et al., 2012*; *Klein Geltink and Pearce, 2019*). It was shown as early as the 80 s that peripheral blood mononuclear cells in a 'resting' state consume approximately three to five times more SAM for methylation than other, nonhepatic, tissues. Once activated, their SAM consumption is further increased. High SAM utilization was also observed in cultured growing lymphoblasts (*German et al., 1983*). The proper function of the SAM transmethylation pathway and efficient removal of SAH by Ahcy is necessary for chemotaxis in human monocytes, and neutrophils and for lymphocyte effector function (*Pike and DeMeester, 1988*; *Zimmerman et al., 1978*; *Pike et al., 1978*). More recently, stable isotope labeling techniques have allowed deeper insight into metabolic set up of activated immune cells. It has been shown that LPS-activated macrophages synergistically increase glycolysis, one carbon metabolism, serine synthesis pathway, pentose phosphate pathway, de novo purine synthesis, and SAM transmethylation pathway activity, to induce, and maintain a proinflammatory

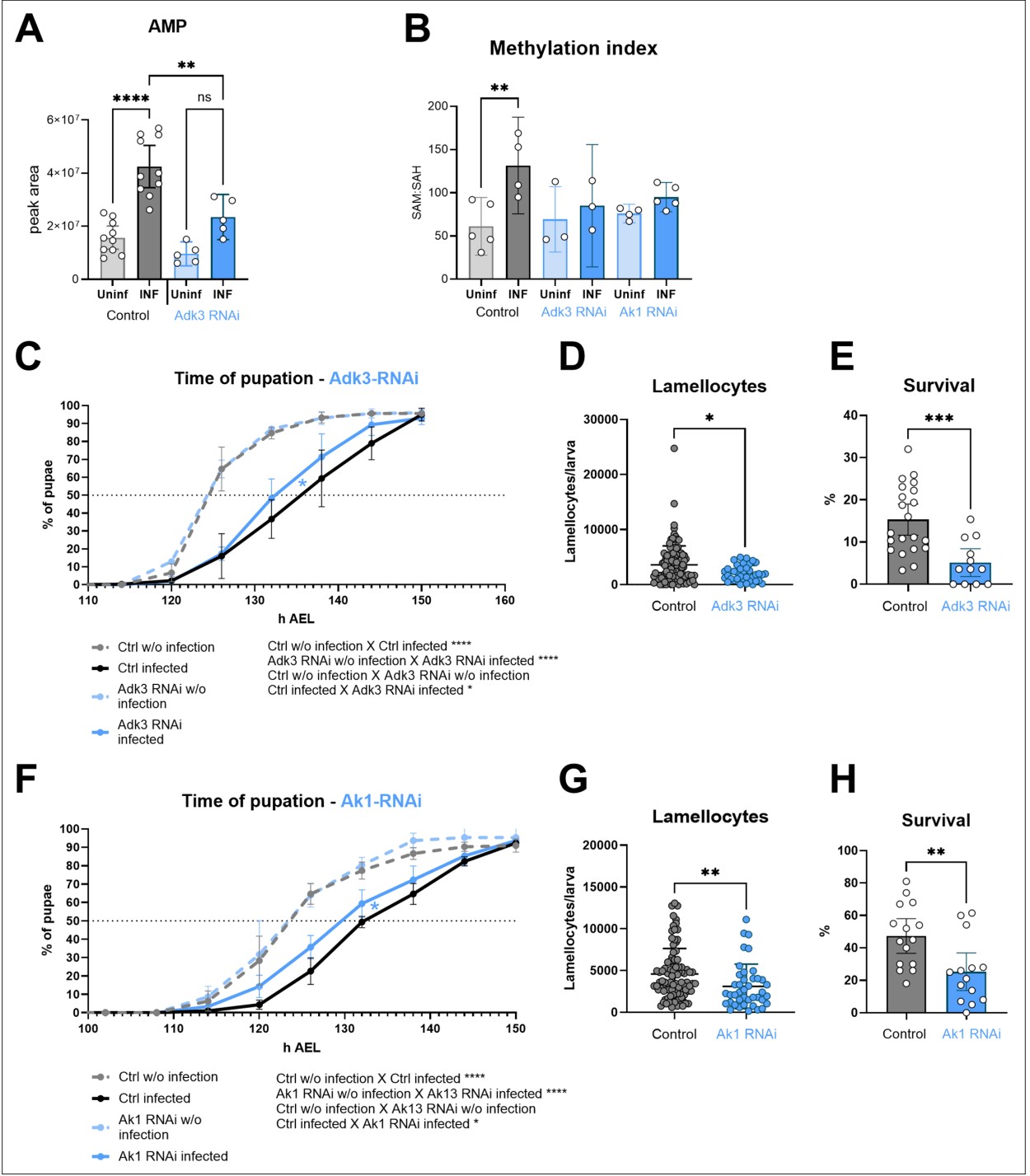

**Figure 5.** Silencing of adenosine kinase and adenylate kinase in hemocytes affects the SAM transmethylation pathway and immune response. (**A**) Total levels of AMP in hemocytes shown as the mean metabolite amounts expressed by the normalized peak area. Infection significantly increases the AMP level in control hemocytes (uninfected - Uninf - light gray vs. infected - INF - dark gray). Hemocyte-specific adenosine kinase Adk3 knockdown (Adk3 RNAi; uninfected - Uninf - light blue and infected - INF - dark blue) significantly decreases AMP during infection compared to control. (**B**) Methylation index, calculated as the ratio of SAM:SAH levels (peak areas; numerical values in *Figure 5—source data 1* and *Supplementary file 1*), in control (uninfected - Uninf - light gray and infected - INF - dark gray) and in hemocyte-specific Adk3-RNAi and adenylate kinase Ak1-RNAi (uninfected - Uninf - light blue and infected - INF - dark blue). (**C**) Pupation is delayed (10 hr) upon infection in control larvae (n=270, uninfected dashed gray line and n=240, infected solid black line) but significantly less (7 hr) in hemocyte-specific Adk3 RNAi larvae (n=275, uninfected dashed blue line and n=275, infected solid blue line). (D) The number of lamellocytes is significantly lower in infected Adk3-RNAi (blue) larvae compared to infected control (gray). (**E**) Percentage of infected larvae surviving to adulthood is significantly lower in Adk3-RNAi (blue) compared to control (gray). (**F**) Pupation is delayed (9 hr) upon

*Figure 5 continued on next page*

*Figure 5 continued*

infection in control larvae (n=225, uninfected dashed gray line and n=270, infected solid black line) but significantly less (6 hr) in hemocyte-specific Ak1 RNAi larvae (n=195, uninfected dashed blue line and n=225, infected solid blue line). (**G**) The number of lamellocytes is significantly lower in infected Ak1-RNAi (blue) larvae compared to infected control (gray). (H) Percentage of infected larvae surviving to adulthood is significantly lower in Ak1-RNAi (blue) compared to control (gray). (**A,B,D,E,G,H**) Bars/lines represent mean values with 95% CI of uninfected (Uninf, light grey) and infected (INF, dark grey) control and uninfected (Uninf, light blue) and infected (INF, dark blue) Adk3-RNAi or Ak1-RNAi samples; each dot represents one biological replicate (numerical values in *Figure 5—source data 1* and *Supplementary file 1*); asterisks represent significant differences between samples tested by unpaired t-test or ordinary one-way ANOVA Tukey's multiple comparison test (*p<0.05, **p<0.01, ***p<0.001, ****p<0.0001). (**C,F**) Lines represent percentages of pupae at hours after egg laying (h AEL); rates were compared using Log-rank survival analysis.

The online version of this article includes the following source data and figure supplement(s) for figure 5:

**Source data 1.** The MS Excel file containing the raw data/numerical values used to generate the plots in the figure.

**Figure supplement 1.** Adenosine kinase Adk3 and adenylate kinase Ak1 knockdown efficiency in hemocytes.

**Figure supplement 1—source data 1.** The MS Excel file containing the raw data/numerical values used to generate the plots in the figure.

phenotype. Interestingly, the homocysteine produced by these macrophages is not recycled back into methionine, but SAM synthesis is likely fed exclusively with exogenous methionine transported into the cells (*Yu et al., 2019*). *Roy et al., 2020* demonstrated the metabolic changes that occur in activated CD4 +and CD8+T lymphocytes. They identified methionine as a crucial metabolite in driving T cells proinflammatory phenotype via epigenetic reprogramming and demonstrated dietary methionine restriction as an effective intervention for autoimmune disease. Activated T lymphocytes, like macrophages, upregulate glycolysis, serine, glycine, and one-carbon metabolism including the SAM transmethylation pathway. These studies indicated that the folate cycle and the SAM transmethylation pathway are uncoupled and homocysteine is not remethylated. Importantly, SAM is synthesized from exogenous methionine and ATP, predominantly produced by de novo purine synthesis. Here, we show that in *Drosophila* immune cells, SAM is also synthesized mostly from exogenous methionine, and homocysteine is not remethylated back to methionine, but rather used in transsulfuration pathway. Interestingly, our results also suggest an alternative means of CTH production from methionine, independent of Ahcy, although this is not further explored here. Activated hemocytes also increase de novo purine synthesis, supported by increased pentose phosphate pathway activity (*Kazek et al., 2024*).

Ahcy was shown to play an important role in adenosine production in heart muscle cells, under normoxic conditions (*Lloyd et al., 1988*; *Deussen et al., 1989*). Adenosine produced by Ahcy contributes to viability of hepatocellular carcinoma cells (HepG2), since Ahcy knockdown followed by adenosine depletion leads to activation of the DNA damage response, cell cycle arrest, a decreased proliferation rate and DNA damage (*Belužić et al., 2018*). However, the adenosine formed in the SAM transmethylation pathway is usually referred to in studies as a by-product that needs to be removed quickly to avoid Ahcy reverse function, which would block further methylation. Here, we show that adenosine formed in the SAM transmethylation pathway is not a mere by-product but can have a significant impact on the immune response and the physiology of the whole organism. With the increase of SAM transmethylation pathway activity in activated hemocytes, more adenosine is produced. As intracellularly adenosine decreases, it is released to the extracellular space; this leads to suppression of metabolism in non-immune tissues, and consequently a development delay. This metabolic suppression is crucial for redirecting nutrients towards the immune system and mounting an effective immune response (*Bajgar et al., 2015*). Knockdown of Ahcy by RNAi suppresses the infection-induced increase in extracellular adenosine, as well as the delay in development, demonstrating that this important systemic signal is most likely generated by the SAM transmethylation pathway. It is important to emphasize that we measured adenosine release only ex vivo (short incubation of hemocytes obtained by larval bleeding), because it is extremely challenging to analyze extracellular adenosine levels in vivo due to the rapid removal of this molecule. Of course, the in vivo situation may be different. We have only been able to detect changes in adenosine levels in vivo in a null mutant for the secreted adenosine deaminase ADGF-A (*Dolezal et al., 2005*) and therefore we rely on genetic manipulations in in vivo studies. Using such manipulations, we have shown in previous work that it is activated hemocytes that secrete adenosine as a signal leading to developmental delay (*Bajgar et al., 2015*). Simply measuring the increase in SAM transmethylation pathway activity in hemocytes, which inevitably leads to increased adenosine production, shows that this pathway at least

contributes to the production of this extracellular signal. This is then strongly supported by the phenotypes obtained here using Ahcy-RNAi, which correspond to phenotypes observed after adenosine manipulations (*Bajgar et al., 2015*).

At the same time, we do not want to suggest that SAM transmethylation in hemocytes is the only source of adenosine during infection. Ectoenzymes are an important source of extracellular adenosine that convert extracellular ATP to ADP, AMP, and adenosine. This cascade is an important immuno-modulator in mammals (*Antonioli et al., 2022*). *Drosophila* also has this cascade of ectoenzymes (*Fenckova et al., 2011*), which appears not to have an immunomodulatory role, likely due to more rudimentary adenosine signaling (*Dolezelova et al., 2007*), but could still serve as a source of extracellular adenosine.

It is also important to add that while adenosine-mediated metabolic suppression and developmental delay are important for an effective immune response, it is likely that the ineffective immune response in Ahcy RNAi animals is at least in part due to suppression of the SAM transmethylation pathway itself in hemocytes, which is required for their function. Ahcy RNAi not only suppresses the production of adenosine, but also homocysteine, an important precursor for the transsulfuration pathway and thus the production of antioxidants and $H_2S$; interestingly, our work suggests an alternative production of homocysteine upon Ahcy RNAi. Ahcy RNAi also leads to extensive accumulation of SAH, so we cannot exclude that some phenotypes are due to this accumulation. Even if accumulated SAH would not cause any harm, it suppresses transmethylation reactions and polyamine synthesis, which certainly affects the function of immune cells (*Latour et al., 2020*; *Xuan et al., 2023*). Therefore, we expect that the effect of Ahcy RNAi on the function of hemocytes is not primarily due to the lack of adenosine as a signal, but rather to the suppression of the SAM transmethylation pathway and its branches. Nevertheless, the increase in the SAM transmethylation pathway in activated hemocytes, leading to increased adenosine production in wild-type hemocytes, and the absence of an increase in extracellular adenosine (at least ex vivo) together with the reduced developmental delay (phenotype obtained by manipulating extracellular adenosine by other means) upon hemocyte-specific knockdown of Ahcy strongly suggest the essential role of the SAM transmethylation pathway in generating adenosine as a systemic signal.

After ATP, SAM is the most used substrate in the cell, with up to half of the daily methionine intake in humans used for SAM synthesis, the vast majority of which is used for methylation (*Lu, 2000*). If all the adenosine produced by Ahcy was released to hemolymph, hemocytes would likely lose a huge amount of ATP due to SAM synthetase utilizing the adenosyl group of ATP and methionine to create SAM. Increased ATP consumption by the accelerating the SAM transmethylation pathway must be supported by de novo purine synthesis. Previous work (*Kazek et al., 2024*) as well as results presented here, show increased de novo purine synthesis in activated immune cells. However, this consumes a considerable amount of energy (six ATP for one IMP molecule) and takes 11 steps to produce IMP from the initial ribose-5-phosphate. Thus, it is more efficient to recycle at least some ATP from the adenosine generated in the SAM transmethylation pathway via coordinated action of adenosine kinase, cytosolic adenylate kinase, and glycolysis. This is in line with ATP synthesis from adenosine via AMP and ADP shown to be significantly increased in activated rat peritoneal macrophages (*Barankiewicz and Cohen, 1985*).

Adenosine kinase has been shown to be coupled with Ahcy to effectively remove adenosine and keep the SAM transmethylation pathway running (*Moffatt et al., 2002*; *Xu et al., 2017*; *Murugan et al., 2021*). Subsequently, the AMP produced can be utilized by the adenylate kinase catalyzing phosphoryl exchange reaction AMP +ATP ↔ 2 ADP. Adenylate kinase is crucial for cellular energy homeostasis, by regulating nucleotides ratio, AMP production and direction to diverse metabolic sensors (e.g. AMPK). In addition, it is responsible for the transfer of ATP-energy-rich phosphoryl between energy producing and energy utilizing sites by a sequence of repeated reactions, creating an effective energy transfer shuttle across the cell (*Zeleznikar et al., 1990*; *Zeleznikar et al., 1995*; *Dzeja et al., 2007*; *Dzeja and Terzic, 2009*). Our hemocyte RNAseq and qPCR data show that *Drosophila* Adk3 and Ak1 are expressed at low levels in quiescent hemocytes, but their expression increases dramatically during infection. Experiments using 13C-labeled adenosine show its incorporation into AMP, ADP, ATP, and especially into SAM, in which the incorporation is significantly increased upon hemocyte activation. These results connect adenosine recycling with the SAM transmethylation pathway. Without infection, the methylation index of Adk3 and Ak1 RNAi in hemocytes is the same

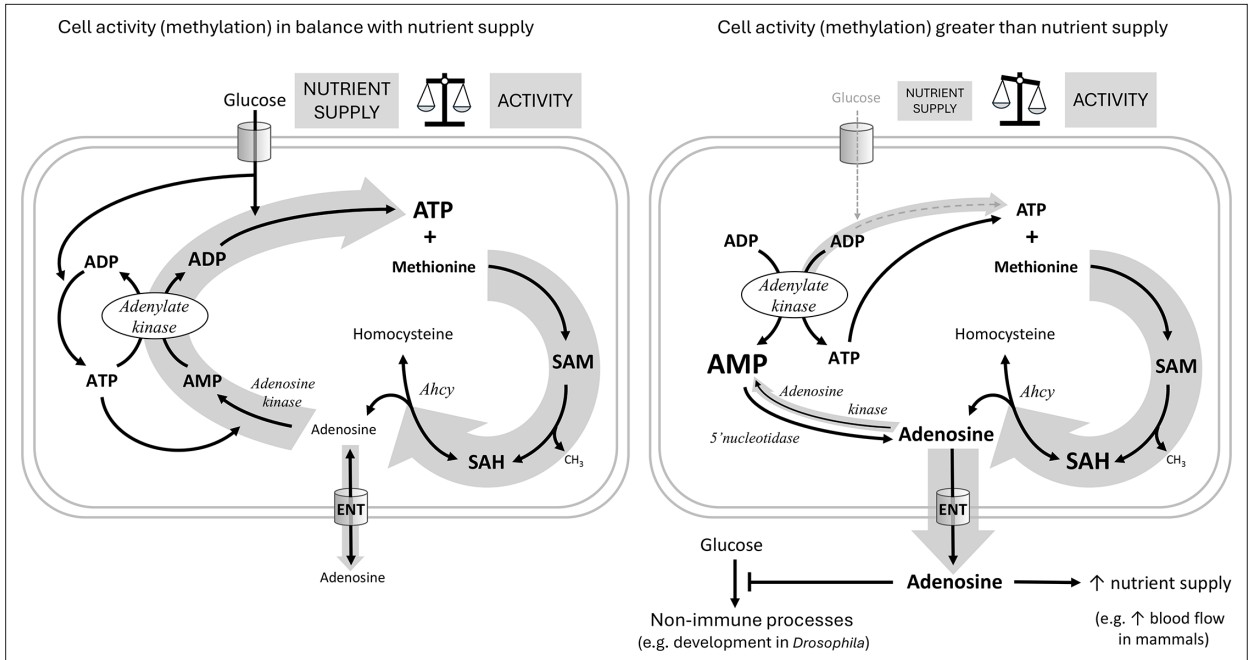

**Figure 6.** Hypothetical role of adenosine as a sensor of the balance between cell activity and nutrient supply. The hypothetical scheme shows an immune cell with either sufficient (left) or insufficient (right) nutrient supply for the given cellular activity. For each individual transmethylation event using S-adenosylmethionine (SAM) as a methyl group donor and producing S-adenosylhomocysteine (SAH), S-adenosylhomocysteinase (Ahcy) generates one molecule of adenosine (and homocysteine). Adenosine thus reflects the sum of all SAM transmethylations in the cell and may be considered as a proxy for cell activity (thick gray arrow on the right). If there are enough nutrients (cell on the left) to maintain high levels of ATP, adenosine can be recycled first to AMP by adenosine kinase, which requires the first ATP, then to ADP by adenylate kinase, which requires the second ATP, and finally to ATP by glycolysis (or oxidative phosphorylation) using glucose (the gray arrow on the left as thick as the one on the right expresses the balance). Recycled ATP can then enter the next round of transmethylation with methionine to form SAM. If there are not enough nutrients (cell on the right, thin gray arrow on the left) to regenerate ATP from ADP, ATP is produced from two ADP molecules by adenylate kinase, simultaneously generating AMP. Accumulated AMP prevents adenosine recycling by adenosine kinase (thin gray arrow on the left), and on the contrary, more adenosine can be produced from AMP by 5'nucleotidase. In this case, adenosine is pushed out of the cell (thick gray arrow down) via the equilibrative nucleoside transporter (ENT) and becomes an extracellular signaling molecule. Extracellular adenosine can, for example, suppress the development of *Drosophila* larvae or increase blood flow in mammals to provide more nutrients to immune cells.

as in control cells. However, upon infection, there is no increase in the methylation index in Adk3 and Ak1 RNAi cells. When the SAM transmethylation pathway is not accelerated, Ahcy does not produce more adenosine, and consequently the adenosine-induced developmental delay during infection is reduced in Adk3 and Ak1 RNAi, whereas without infection, RNAi larvae develop at the same rate as controls. Our results therefore demonstrate a novel functional connection of adenosine recycling to ATP with SAM transmethylation pathway. At least under energy demanding conditions, as during the activation of immune cells, Adk and cytosolic Ak activities appear to support SAM transmethylation pathway activity.

Similar to Ahcy RNAi, Adk3 and Ak1 RNAi in hemocytes also negatively affects immune response efficiency. We see a significant decrease in lamellocytes number in our infected knockdown larvae, resulting in decreased survival. It is important to note, that while Ak1 RNAi most likely also affects de novo purine synthesis, the Adk3 RNAi effects we observed are due to impairments in adenosine recycling. Connecting the SAM transmethylation pathway-driven production of adenosine with its recycling via Adk3 and Ak1 leads us to following hypothesis (*Figure 6*):

Most cellular processes involve methylation, therefore the SAM transmethylation pathway reflects the activity of the cell. While a wide variety of molecules are methylated, adenosine is the universal product of each individual methylation. Thus, adenosine production reflects the sum of all methylations (where SAM has been used) and therefore the overall activity of the cell. Adenosine must be removed quickly as to not block the SAM transmethylation pathway. The preferred fate is to recycle adenosine to AMP and eventually ATP so that the cell does not lose its adenosyl pool. However, this requires nutrients, adenosine kinase requires ATP, adenylate kinase produces ADP from AMP and ATP

and the resulting ADP is recycled to ATP in glycolysis, consuming glucose. If the cell's nutrient supply is inadequate, AMP begins to accumulate, due to the opposite action of adenylate kinase, preventing adenosine from being recycled to AMP by adenosine kinase. In this case, the accumulating adenosine is pushed out of the cell via ENT, where it becomes a signaling molecule, informing surrounding tissues of the need for more nutrients. We therefore look at adenosine as a sensitive sensor of the balance between cell activity (reflected in methylation) and nutrient supply – the better the balance, the less adenosine is released and the more is recycled.

To experimentally test this hypothesis will require extremely well-controlled conditions, as offered for example by uniform cell culture in precisely controlled media, and delicate manipulation of the important players. The proposed role of adenosine as a sensitive sensor between cell activity and nutrient supply might have far-reaching implications. The production of adenosine by nucleotidase as the ADP/ATP ratio rises is known, for example, to generate a signal in ischemic tissue to inform surrounding tissues of metabolic stress (e.g. vasodilation to increase blood flow). However, the generation of such a signal as a result of an imbalance between cell activity and nutrient supply is a novel and exciting hypothesis.

The fate of adenosine is determined by biochemical properties and the expression of enzymes and transporters. The Km values of *Drosophila* Adk2 and Adk3 are not known (*Fleischmannova et al., 2012*; *Stenesen et al., 2013*), but it is possible that they differ, and that increased expression of Adk3 specifically during infection is important in readjusting the balances between AMP and adenosine, which are determined by the properties of nucleotidases and adenosine kinases (*Boison, 2013*). The expression levels and properties of nucleoside transporters and adenosine deaminase also play important roles. The specific setup of this whole system in certain cell types under certain conditions determines what happens with adenosine (*Dulla and Masino, 2013*; *Dale, 2021*). Studying this setup is essential to understanding the role of this system in specific situations, such as those in which adenosine may be generated by imbalances in activity and energy supply. For example, does it play a role in the early activation of immune cells, regulating nutrient supply to their site of activation? In addition, in the tumor microenvironment, adenosine inhibits infiltrating immune cells (*Boison and Yegutkin, 2019*). Until now adenosine ectoenzymes have been mainly studied in this respect. However, the SAM transmethylation pathway is very active in tumor cells (*Vizán et al., 2021*) – can tumors regulate their environment by releasing adenosine, which is produced according to our hypothetical mechanism? Basically, any cell/tissue in which the SAM transmethylation pathway is active can potentially use adenosine as a sensor of the balance between its activity and nutrient supply. Some tissues will probably be more privileged and demand nutrients by releasing adenosine because they have more nucleoside transporters expressed. For example, in the brain, the transmethylation pathway is important for many different processes (*Jin et al., 2018*), and extracellular adenosine is an important neuromodulator and regulator of blood flow (*Peng et al., 2020*). Less privileged tissues with lower expression of nucleoside transporters may be more likely to convert adenosine to AMP and activate AMPK during nutrient deprivation to dampen their overactivity. These speculations, of course, need to be tested in the future.

## Methods

**Key resources table**

| Reagent type (species) or resource | Designation | Source or reference | Identifiers | Additional information |
|---|---|---|---|---|
| Gene (*Drosophila melanogaster*) | *Ahcy* | GenBank | FLYB: FBgn0014455 | |
| Gene (*Drosophila melanogaster*) | *Ak1* | GenBank | FLYB: FBgn0022709 | |
| Gene (*Drosophila melanogaster*) | *Adk3* | GenBank | FLYB: FBgn0026602 | |

*Continued on next page*

*Continued*

| Reagent type (species) or resource | Designation | Source or reference | Identifiers | Additional information |
|---|---|---|---|---|
| Genetic reagent (*Drosophila melanogaster*) | UAS-Ahcy-RNAi | Bloomington *Drosophila* Stock Center | BDSC: 67848 FLYB: FBti0186788 | |
| Genetic reagent (*Drosophila melanogaster*) | UAS-Adk3-RNAi | Bloomington *Drosophila* Stock Center | BDSC: 67755 FLYB: FBti0185498 | |
| Genetic reagent (*Drosophila melanogaster*) | UAS-Adk2-RNAi | Bloomington *Drosophila* Stock Center | BDSC: 35167 FLYB: FBti0144142 | |
| Genetic reagent (*Drosophila melanogaster*) | UAS-Ak1-RNAi | Bloomington *Drosophila* Stock Center | BDSC: 35582 FLYB: FBti0144266 | |
| Genetic reagent (*Drosophila melanogaster*) | P{CaryP}attP2 | Bloomington *Drosophila* Stock Center | BDSC: 36303 FLYB: FBst0036303 | |
| Genetic reagent (*Drosophila melanogaster*) | P{CaryP}Msp300attP40 | Bloomington *Drosophila* Stock Center | BDSC: 36304 FLYB: FBst0036304 | |
| Genetic reagent (*Drosophila melanogaster*) | *SrpD-Gal4* | Crozatier | FLYB: FBtp0020112 | |
| Genetic reagent (*Drosophila melanogaster*) | P{tubP-GAL80ts}2 | Bloomington *Drosophila* Stock Center | BDSC: 7017 FLYB: FBti0027797 | |
| Biological sample (*Leptopilina boulardi*) | *Leptopilina boulardi* | Crozatier | NCBI:txid63433, RRID:NCBITaxon_63433 | |
| Sequence-based reagent | Ahcy Fw | This paper | PCR primers | AAGCTGTCGCACAAATGGCG |
| Sequence-based reagent | Ahcy Rev | This paper | PCR primers | GCACGTTGTGCACCAGGAAC |
| Sequence-based reagent | AhcyL1 Fw | This paper | PCR primers | GGCGAGACGGAAGAGGACT |
| Sequence-based reagent | AhcyL1 rev | This paper | PCR primers | AGAGAGCTGATAGAGACGGTG |
| Sequence-based reagent | Adk2 Fw | This paper | PCR primers | AACTTCACCATCGATCACCTGG |
| Sequence-based reagent | Adk2 Rev | This paper | PCR primers | TAGGAACGGTCGCTGTTTGG |
| Sequence-based reagent | Adk3 Fw | This paper | PCR primers | CTGGACATGGAGAAACTCAACC |
| Sequence-based reagent | Adk3 Rev | This paper | PCR primers | AAAGAAGAGCGCGTCTGTGC |
| Sequence-based reagent | Ak1 Fw | This paper | PCR primers | CTCGGCATTGATCGTAAGGG |
| Sequence-based reagent | Ak1 Rev | This paper | PCR primers | CGATCTGGCGCTGTACTTTG |
| Sequence-based reagent | Ak2 Fw | This paper | PCR primers | GCTGAGAAGCTCGACACATTG |
| Sequence-based reagent | Ak2 Rev | This paper | PCR primers | CTCCAGTGACATCGTCCGTC |

*Continued on next page*

*Continued*

| Reagent type (species) or resource | Designation | Source or reference | Identifiers | Additional information |
|---|---|---|---|---|
| Sequence-based reagent | Ak3 Fw | This paper | PCR primers | GATCCAGCGATTCTTGACCC |
| Sequence-based reagent | Ak3 Rev | This paper | PCR primers | CGGATAACCGAGGTAGGCAAC |
| Sequence-based reagent | ArgK Fw | This paper | PCR primers | ATGGAGATGATGCGGAGATG |
| Sequence-based reagent | ArgK Rev | This paper | PCR primers | TCGACGACCACTTCCTGTTC |
| Sequence-based reagent | Cbs Fw | This paper | PCR primers | AGATTACGCCCAACATCCTCG |
| Sequence-based reagent | Cbs Rev | This paper | PCR primers | AATGCGGTCCTTCACTGATCC |
| Sequence-based reagent | CG10621 Fw | This paper | PCR primers | CATCGAGCTGATAAAGAACACGG |
| Sequence-based reagent | CG10621 Rev | This paper | PCR primers | AATGGAGGCAATGATCAAAGGG |
| Sequence-based reagent | CG20623 Fw | This paper | PCR primers | ATGCCAAATTCTGGGTCTCCC |
| Sequence-based reagent | CG20623 Rev | This paper | PCR primers | AAAGGGGTCACGAATAGTGGG |
| Sequence-based reagent | RpL32 Fw | This paper | PCR primers | AAGCTGTCGCACAAATGGCG |
| Sequence-based reagent | RpL32 Rev | This paper | PCR primers | GCACGTTGTGCACCAGGAAC |
| Sequence-based reagent | SamDC Fw | This paper | PCR primers | CAACGGTGACGATGATCTGC |
| Sequence-based reagent | SamDC Rev | This paper | PCR primers | AGTTTTAAGGATCCATCGTCGC |
| Sequence-based reagent | SamS Fw | This paper | PCR primers | CAAATCAGCGACGCTATCTTGG |
| Sequence-based reagent | SamS Rev | This paper | PCR primers | TGTCTCACGAACAACCTTCTGG |
| Chemical compound | L-methionine-$^{13}C_5$ | Sigma-Aldrich | 908339 | |
| Chemical compound | adenosine-$^{13}C_5$ | Cambridge Isotope Laboratories | CLM-3678 | |

## Fly strains, cultivation, and parazitoid wasp infection

*Drosophila melanogaster* strain $w^{1118}$ (FBal0018186) in Canton S genetic background (FBst0064349) was used as a control line unless otherwise stated. Strains *UAS-Ahcy-RNAi Ahcy^HMS05799^* (FBti0186788), UAS-Adk3-RNAi *Adk3^HMC06354^* (FBti0185498), UAS-Adk2-RNAi *Adk2^GL00036^* (FBti0144142), UAS-Ak1-RNAi *Ak1^GL00177^* (FBti0144266), and control lines for RNAi *y^1 v^1; P{CaryP}attP2* (FBst0036303), and *y^1 v^1; P{CaryP}Msp300attP40* (FBst0036304) were obtained from the Bloomington *Drosophila* Stock Center. The *SrpD-Gal4* strain (FBtp0020112) was obtained from M. Crozatier, backcrossed into the $w^{1118}$ background, and recombined with P{tubP-GAL80ts}2 (FBti0027797), which was also backcrossed into $w^{1118}$ background, to generate the $w^{1118}$; +/+; *SrpD-Gal4* P{tubP-GAL80ts}2 line with Gal4 expression in all hemocytes but not in the fat body when kept at 18 °C for the first 3 days of development and then transferred to 25 °C (S Fig; expression in the fat body is only present at 29 °C in this line). All flies were grown on cornmeal medium (8% cornmeal, 5% glucose, 4% yeast, 1% agar, 0.16% methylparaben) at 25 °C. Parasitoid wasps *Leptopilina boulardi* were reared on sugar agar medium (6% sucrose, 1.5% agar, 0.75% methylparaben) and grown by infection of wild-type *Drosophila* larvae. Early third instar larvae (72 hr after egg laying) were infected with parasitoid wasps (=time point 0 hr). Weak infection

(1–2 eggs per larva) was used for survival analysis. Strong infection (4–8 eggs per larva) was used for the rest of the experiments to obtain a strong and more uniform immune response. Infections were performed on 60 mm Petri dishes with standard cornmeal medium for 15 min for weak infection and 45 min for strong infection.

### Hemocyte counting

*Drosophila* circulating hemocytes were counted at 20 hpi by bleeding one larva in 15 µL of PBS and using Neubauer hemocytometer (*Brand GMBH*) using differential interference contrast microscopy. Particular types of hemocytes were determined based on the cell morphology and since we were mainly interested in lamellocytes, we counted them 22–24 hr after infection, when most of the lamellocytes from the first wave are fully differentiated but still mostly in circulation, as they are just starting to adhere to the wasp egg.

### Survival and developmental time analysis

Infected and non-infected 3rd instar larvae were placed into fresh vials in a number of 30 larvae per vial, three vials per genotype with three to five independent experiments. Differences in larval development were determined by counting newly appeared pupae every 6 hr post infection. Percentage of pupae number per total used larvae number at particular time points was plotted and compared by Log-rank test. For survival, all emerged adult flies were counted and checked if they contain the melanized eggs to confirm they were infected. The ones without any egg were excluded from the analysis. Percentage of emerged adults number per total used infected individuals were plotted.

### Gene expression analysis

Based on the larval stage, 30–50 larvae were washed 3 x in PBS, transferred on microscope slide covered with parafilm and placed on ice. All larvae were opened with tweezers and released hemolymph with hemocytes were aspired with 10 µL pipette and transferred into 1.5 mL tube containing TRIzol Reagent (*Life Technologies*) for RNA extraction. RNA was purified with Direct-zol RNA Microprep (*Zymo-Research*). Total RNA was used for cDNA preparation by PrimeScript Revrese Transcriptase (*Takara*) with oligo(dT) primer. cDNA was used as a template for qPCR using TP SYBR 2 x Master Mix (*Top-Bio*) and C1000 Touch Thermal Cycler (*Bio-Rad*). Expression of a specific gene in each sample was normalized to expression of *RpL32* (FBgn0002626) and relative gene expression was counted with ΔΔCt method. Graphs showing target gene expression and statistical analysis were made with GraphPad Prism.

### Bulk RNAseq analysis

RNA was extracted from circulating hemocytes (72 hr after egg laying = time of infection = 0 hr, 81 hr after egg laying = 9 hr post infection/hpi and 90 hr after egg laying = 18 hpi) of uninfected and infected 3rd instar *w1118* larvae (6–10 replicates) and barcoded 3'-end seq forward libraries were sequenced by deep uni-directional sequencing of 75-base long reads using Illumina NextSeq as described in details in *Kazek et al., 2024*. Raw data are available at The European Nucleotide Archive under study accession number: PRJEB74490 (secondary acc: ERP159178; https://www.ebi.ac.uk/ena/browser/view/PRJEB74490). Trimmed reads were mapped to the BDGP *Drosophila melanogaster* Release 6.29 genomic sequence using the Mapper for RNA Seq in Geneious prime software (Biomatters). Normalized counts of reads mapped to each gene annotation were calculated as transcripts per million (TPM), expression levels were compared using the DESeq2 method in Geneious prime software, and data were exported to an Excel file (S4 Data).

### Metabolomics and stable 13C isotope tracing in hemocytes ex vivo

Samples for metabolomics were collected 20 hours post infection. Larvae were washed first with distilled water and then with PBS to reduce contamination. Larval hemolymph was collected by carefully tearing the larvae on a glass microscope slide covered with parafilm. Hemolymph from 100 larvae was immediately collected into sterile 1.5 ml Eppendorf polypropylene centrifuge tubes prefilled with 100 µl PBS and centrifuged for 5 min at 25 °C, 360 x *g*. For experiments without 13 C isotope tracing, hemocytes were resuspended in 50 µL of HPLC water and three times frozen in liquid nitrogen/thawed at 37 °C (established for sufficient cell disruption and extraction of intracellular metabolites). 200 µl

of cold acetonitrile-methanol (1:1) was added and stored at –80 °C until LC-HRMS analysis described in *Kazek et al., 2024*. For experiments with 13 C isotope tracing, the supernatant was removed and the cells were mixed with medium containing PBS, 5 mM trehalose, 0.5 mM glucose, 5 mM proline, 5 mM glutamine (*Sigma/MERCK*), supplemented with gentamicin (10 mg/ml; *Gibco*), amphotericin B (250 µg/ml; *Gibco*) and 0.1 mM phenylthiourea (PTU; *Sigma/MERCK*) to prevent melanization. Based on the experiment, medium further contained either 0.33 µM L-methionine-$^{13}C_5$ or 10 µM adenosine-$^{13}C_5$ and unlabeled 0.33 µM L-methionine. The concentrations are based on measurements of actual hemolymph concentrations in wild-type larvae in the case of methionine, and in the case of adenosine, we used a slightly higher concentration than measured in the *adgf-a* mutant (*Dolezal et al., 2005*) to have a sufficiently high concentration to allow adenosine to flow into the hemocytes.

The hemocytes were then incubated ex vivo for 20 min at 25 °C and 80–90% humidity. The cells were then centrifuged for 5 min at 25 °C, 360 x *g*, the supernatant was removed, the cells were mixed with 50 µl of cold PBS and frozen in liquid nitrogen/thawed at 37 °C three times (to disrupt the rigid hemocyte cells). Finally, 200 µl of cold acetonitrile-methanol (1:1) was added, in addition, the internal standard 4-fluorophenylalanine (50 ng/sample) was added to control the extraction efficiency and mass spectrometer responses and samples were stored at –80 °C until LC-HRMS analysis which is described in *Kazek et al., 2024*. When extracellular adenosine and inosine were measured, samples were collected in room temperature and centrifuged in 25 °C, 5 min, 360 x *g* to wash hemocytes from hemolymph and extracellular metabolites. Cells were resuspended in medium composed of PBS, 5 mM trehalose, 0.5 mM glucose, 5 mM proline, 5 mM glutamine, 0.33 mM methionine (*Sigma/MERCK*), supplemented with gentamicin (10 mg/ml; *Gibco*), amphotericin B (250 µg/ml; *Gibco*), 0.15 mM EHNA (*Sigma/MERCK*) and 0.1 mM phenylthiourea (PTU; *Sigma/MERCK*) to prevent melanization. Samples were incubated for 20 min in room temperature, and then centrifuged in 4 °C, 5 min, 360 x *g*. 70 µL of supernatant were transferred into new sterile 1.5 mL Eppendorf polypropylene centrifuge tube containing 70 µL of acetonitrile and stored at –80 °C until LC-HRMS analysis. Pelleted cells were stored in –80 °C for protein measurement with QuantiPro BCA Assay Kit (*Sigma/MERCK*).

Raw data are available at https://doi.org/10.6084/m9.figshare.27291300.v1 and processed data in *Supplementary file 1*. For peak area analysis, the data were normalized to the total content of all screened unlabeled metabolites - the peak area of the metabolite in a particular sample was divided by the peak area of the same metabolite of the selected reference sample and this procedure was repeated for each individual unlabeled metabolite. These ratios of all metabolites in one particular sample were averaged to determine a normalization factor. We then divided the measured peak area by the normalization factor for that sample to obtain the normalized peak area values (normalization factors are reported in *Supplementary file 1*).

## Data analysis

Data were analyzed and graphed using GraphPad Prism (*GraphPad Software*), with specific statistical tests shown in the legend of each figure.

## Acknowledgements

We thank Dr. Michele Crozatier and Bloomington *Drosophila* Stock Center for fly and wasp stocks. We thank Dr. Michalina Kazek for help with metabolomics optimization, to Lucie Hrádková for laboratory management and Marcela Jungwirthová for project management, and all members of Doležal and Moos laboratories for their help with work. We thank Dr. Adam Bajgar for help with bulk RNAseq and Dr. Vladimír Beneš and Genomics Core Facility (EMBL Heidelberg, Germany) for RNAseq services and Dr. Bajgar and Dr. Gabriela Krejčová for help with optimization of the molecular biology methods and wasp infections and general manipulation with flies and larvae. We thank Dr. Ellen McMullen for manuscript editing.

## Additional information

### Funding

| Funder | Grant reference number | Author |
|---|---|---|
| Grantová Agentura České Republiky | 17-16406S | Tomáš Doležal |
| Grantová Agentura České Republiky | 20-09103S | Tomáš Doležal |
| Jihočeská Univerzita v Českých Budějovicích | GAJU 087/2019/P | Pavla Nedbalová |

The funders had no role in study design, data collection and interpretation, or the decision to submit the work for publication.

### Author contributions

Pavla Nedbalová, Conceptualization, Formal analysis, Funding acquisition, Investigation, Visualization, Methodology, Writing – original draft, Writing – review and editing; Nikola Kaislerova, Investigation, Visualization; Lenka Chodakova, Investigation, Visualization, Writing – review and editing; Martin Moos, Conceptualization, Resources, Data curation, Formal analysis, Supervision, Validation, Investigation, Visualization, Methodology, Writing – review and editing; Tomáš Doležal, Conceptualization, Resources, Data curation, Formal analysis, Supervision, Funding acquisition, Investigation, Visualization, Methodology, Writing – original draft, Project administration, Writing – review and editing

### Author ORCIDs

Tomáš Doležal ⓘ https://orcid.org/0000-0001-5217-4465

Reviewer #1 (Public review): https://doi.org/10.7554/eLife.105039.3.sa1
Reviewer #2 (Public review): https://doi.org/10.7554/eLife.105039.3.sa2
Reviewer #3 (Public review): https://doi.org/10.7554/eLife.105039.3.sa3
Author response https://doi.org/10.7554/eLife.105039.3.sa4

## Additional files

### Supplementary files

Supplementary file 1. Metabolomics and stable $^{13}$C isotope tracing in circulating hemocytes during parasitoid wasp infection. MS Excel sheets with stable $^{13}$C isotope tracing experiments. List of metabolites, their characterizations, identification, HPLC/HRMS parameters is on the first sheet [List of Metabolites]. Data from the following experiments are in individual sheets (values are raw or normalized areas under respective chromatographic peaks): [S1] Peak area of metabolites of interest for control and Ahcy RNAi hemocytes. [S2] – Combination of quantitative analysis (ng/sample) and peak area of metabolites of interest for control and Ahcy RNAi hemocytes incubated with 13C5 methionine. [S3] Peak area of metabolites of interest for control and Adk3 RNAi and Ak1 RNAi. [S4] Combination of quantitative analysis (ng/sample) and peak area of metabolites of interest for control and Adk3 RNAi and Ak1 RNAi hemocytes incubated with 13C5 adenosine. Raw data are available at figshare (https://doi.org/10.6084/m9.figshare.27291300.v1).

Supplementary file 2. Methyltransferases gene expression analysis by bulk RNAseq of circulating hemocytes. MS Excel sheets with expression of genes selected from *Supplementary file 3*, representing methyltransferases according to Flybase database.

Supplementary file 3. Gene expression analysis by bulk RNAseq of circulating hemocytes. MS Excel sheets with gene expression in circulating hemocytes. RNA was extracted 72 hours after (egg laying = time of infection = 0 hours, 81 hours after egg laying = 9 hours post infection/hpi and 90 hours after egg laying = 18 hpi), from hemocytes of the third instar $w^{1118}$ larvae. This table was published in *Kazek et al., 2024*. Raw data are available at The European Nucleotide Archive under Study accession number: PRJEB74490 (secondary acc: ERP159178) (https://www.ebi.ac.uk/ena/browser/view/PRJEB74490).

MDAR checklist

## Data availability

Raw data of metabolomics and stable 13C isotope tracing in circulating hemocytes during parasitoid wasp infection have been deposited to figshare and are available at https://doi.org/10.6084/m9.figshare.27291300.v1. RNAseq raw data are available at The European Nucleotide Archive under study accession number: PRJEB74490 (secondary acc: ERP159178) (https://www.ebi.ac.uk/ena/browser/view/PRJEB74490). All data generated or analysed during this study are included in the manuscript and supporting files; source data files have been provided for Figures 2–5.

The following datasets were generated:

| Author(s) | Year | Dataset title | Dataset URL | Database and Identifier |
|---|---|---|---|---|
| Kazek M, Chodáková L, Lehr K, Strych L, Nedbalová P, McMullen E, Bajgar A, Opekar S, Šimek P, Moos M, Doležal T | 2024 | Glucose and trehalose metabolism through the cyclic pentose phosphate pathway shape pathogen resistance and host protection in *Drosophila* | https://www.ebi.ac.uk/ena/browser/view/PRJEB74490 | European Nucleotide Archive, PRJEB74490 |
| Doležal T, Nedbalová P, Kaislerova N, Chodakova L, Moos M | 2024 | Metabolomic data | https://doi.org/10.6084/m9.figshare.27291300.v1 | figshare, 10.6084/m9.figshare.27291300.v1 |

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
