## [Editor Report · eLife Assessment]

This paper provides a **valuable** contribution to our understanding of how adenosine acts as a signal of nutrient insufficiency and extends this idea to suggest that adenosine is released by metabolically active cells in proportion to the activity of methylation events. **Convincing** data supports this idea. The authors use metabolic tracing approaches to identify the biochemical pathways that contribute to the regulation of adenosine levels and the S-adenosylmethionine cycle in *Drosophila* larval hemocytes in response to wasp egg infection.

---

## [Referee Report · Reviewer #1 (Public review)]

Summary:

In this article, Nedbalova et al. investigate the biochemical pathway that acts in circulating immune cells to generate adenosine, a systemic signal that directs nutrients toward the immune response, and S-adenosylmethionine (SAM), a methyl donor for lipid, DNA, RNA, and protein synthetic reactions. They find that SAM is largely generated through uptake of extracellular methionine, but that recycling of adenosine to form ATP contributes a small but important quantity of SAM in immune cells during the immune response. The authors propose that adenosine serves as a sensor of cell activity and nutrient supply, with adenosine secretion dominating in response to increased cellular activity. Their findings of impaired immune action but rescued larval developmental delay when the enzyme Ahcy is knocked down in hemocytes are interpreted as due to effects on methylation processes in hemocytes and reduced production of adenosine to regulate systemic metabolism and development, respectively. Overall this is a strong paper that uses sophisticated metabolic techniques to map the biochemical regulation of an important systemic mediator, highlighting the importance of maintaining appropriate metabolite levels in driving immune cell biology.

Strengths:

The authors deploy metabolic tracing - no easy feat in *Drosophila* hemocytes - to assess flux into pools of the SAM cycle. This is complemented by mass spectrometry analysis of total levels of SAM cycle metabolites to provide a clear picture of this metabolic pathway in resting and activated immune cells.

The experiments show that recycling of adenosine to ATP, and ultimately SAM, contributes meaningfully to the ability of immune cells to control infection with wasp eggs.

This is a well-written paper, with very nice figures showing metabolic pathways under investigation. In particular, the italicized annotations, for example "must be kept low", in Figure 1 illustrate a key point in metabolism - that cells must control levels of various intermediates to keep metabolic pathways moving in a beneficial direction.

Experiments are conducted and controlled well, reagents are tested, and findings are robust and support most of the authors' claims.

Weaknesses:

The authors posit that adenosine acts a sensor of cellular activity, with increased release indicating active cellular metabolism and insufficient nutrient supply. The authors have provided a discussion of how generalizable they think this may be across different cell types or organs, but mechanisms for the role of adenosine in specific cell types, and whether cell autonomous or cell-nonautonomous mechanisms may be employed in sensing, are largely unknown.

---

## [Referee Report · Reviewer #2 (Public review)]

Summary:

In this work, the authors wish to explore the metabolic support mechanisms enabling lamellocyte encapsulation, a critical antiparasitic immune response of insects. They show that S-adenosylmethionine metabolism is specifically important in this process through a combination of measurements of metabolite levels and genetic manipulations of this metabolic process.

Strengths:

The metabolite measurements and the functional analyses are generally very strong, and clearly show that the metabolic process under study is important in lamellocyte immune function.

Previous weaknesses:

The previous version of the manuscript contained RNAseq data that were inadequately explained. In this version, the treatment and representation of these data are significantly improved, such that they no longer represent a significant weakness. This version also contains increased evidence that SAM transmethylation is directly required for encapsulation.

---

## [Referee Report · Reviewer #3 (Public review)]

Summary:

The authors of this study provides evidence that *Drosophila* immune cells show upregulated SAM transmethylation pathway and adenosine recycling upon wasp infection. Blocking this pathway compromises the lamellocyte formation, developmental delay and the host survival, suggesting its physiological relevance.

Strengths:

Snapshot quantification of the metabolite pool does not provide evidence that the metabolic pathway is active or not. The authors use an ex vivo isotope labelling to precisely monitor the SAM and adenosine metabolism. During infection, the methionine metabolism and adenosine recycling are upregulated, which is necessary to support the immune reaction. By combining the genetic experiment, they successfully show that the pathway is activated in immune cells.

Weaknesses:

The authors knocked down Ahcy to prove the importance of SAM methylation pathway. However, Ahcy-RNAi produces massive accumulation of SAH, in addition to block adenosine production. To further validate the phenotypic causality, it is important to manipulate other enzymes in the pathway, such as Sam-S, Cbs, SamDC, etc. The authors do not demonstrate how infection stimulates the metabolic pathway given the gene expression of metabolic enzymes is not upregulated by infection stimulus.

---

## [Author Response]

The following is the authors’ response to the original reviews.

**Public Reviews:**

**Reviewer #1 (Public review):**
Summary:In this article, Nedbalova et al. investigate the biochemical pathway that acts in circulating immune cells to generate adenosine, a systemic signal that directs nutrients toward the immune response, and S-adenosylmethionine (SAM), a methyl donor for lipid, DNA, RNA, and protein synthetic reactions. They find that SAM is largely generated through the uptake of extracellular methionine, but that recycling of adenosine to form ATP contributes a small but important quantity of SAM in immune cells during the immune response. The authors propose that adenosine serves as a sensor of cell activity and nutrient supply, with adenosine secretion dominating in response to increased cellular activity. Their findings of impaired immune action but rescued larval developmental delay when the enzyme Ahcy is knocked down in hemocytes are interpreted as due to effects on methylation processes in hemocytes and reduced production of adenosine to regulate systemic metabolism and development, respectively. Overall this is a strong paper that uses sophisticated metabolic techniques to map the biochemical regulation of an important systemic mediator, highlighting the importance of maintaining appropriate metabolite levels in driving immune cell biology.Strengths:The authors deploy metabolic tracing - no easy feat in *Drosophila* hemocytes - to assess flux into pools of the SAM cycle. This is complemented by mass spectrometry analysis of total levels of SAM cycle metabolites to provide a clear picture of this metabolic pathway in resting and activated immune cells.The experiments show that the recycling of adenosine to ATP, and ultimately SAM, contributes meaningfully to the ability of immune cells to control infection with wasp eggs.This is a well-written paper, with very nice figures showing metabolic pathways under investigation. In particular, the italicized annotations, for example, "must be kept low", in Figure 1 illustrate a key point in metabolism - that cells must control levels of various intermediates to keep metabolic pathways moving in a beneficial direction.Experiments are conducted and controlled well, reagents are tested, and findings are robust and support most of the authors' claims.Weaknesses:The authors posit that adenosine acts as a sensor of cellular activity, with increased release indicating active cellular metabolism and insufficient nutrient supply. It is unclear how generalizable they think this may be across different cell types or organs.

In the final part of the Discussion, we elaborate slightly more on a possible generalization of our results, while being aware of the limited space in this experimental paper and therefore intend to address this in more detail and comprehensively in a subsequent perspective article.

The authors extrapolate the findings in Figure 3 of decreased extracellular adenosine in ex vivo cultures of hemocytes with knockdown of Ahcy (panel B) to the in vivo findings of a rescue of larval developmental delay in wasp egg-infected larvae with hemocyte-specific Ahcy RNAi (panel C). This conclusion (discussed in lines 545-547) should be somewhat tempered, as a number of additional metabolic abnormalities characterize Ahcy-knockdown hemocytes, and the in vivo situation may not mimic the ex vivo situation. If adenosine (or inosine) measurements were possible in hemolymph, this would help bolster this idea. However, adenosine at least has a very short half-life.

We agree with the reviewer, and in the 4th paragraph of the Discussion we now discuss more extensively the limitations of our study in relation to ex vivo adenosine measurements and the importance of the SAM pathway on adenosine production.

**Reviewer #2 (Public review):**
Summary:In this work, the authors wish to explore the metabolic support mechanisms enabling lamellocyte encapsulation, a critical antiparasitic immune response of insects. They show that S-adenosylmethionine metabolism is specifically important in this process through a combination of measurements of metabolite levels and genetic manipulations of this metabolic process.Strengths:The metabolite measurements and the functional analyses are generally very strong and clearly show that the metabolic process under study is important in lamellocyte immune function.Weaknesses:The gene expression data are a potential weakness. Not enough is explained about how the RNAseq experiments in Figures 2 and 4 were done, and the representation of the data is unclear.

The RNAseq data have already been described in detail in our previous paper (doi.org/10.1371/journal.pbio.3002299), but we agree with the reviewer that we should describe the necessary details again here. The replicate numbers for RNAseq data were added to figure legends, the TPM values for the selected genes shown in figures are in S1_Data and new S4_Data file with complete RNAseq data (TPM and DESeq2) was added to this revised version.

The paper would also be strengthened by the inclusion of some measure of encapsulation effectiveness: the authors show that manipulation of the S-adenosylmethionine pathway in lamellocytes affects the ability of the host to survive infection, but they do not show direct effects on the ability of the host to encapsulate wasp eggs.

The reviewer is correct that wasp egg encapsulation and host survival may be different (the host can encapsulate and kill the wasp egg and still not survive) and we should also include encapsulation efficiency. This is now added to Figure 3D, which shows that encapsulation efficiency is reduced upon Ahcy-RNAi, which is consistent with the reduced number of lamellocytes.

**Reviewer #3 (Public review):**
Summary:The authors of this study provide evidence that *Drosophila* immune cells show upregulated SAM transmethylation pathway and adenosine recycling upon wasp infection. Blocking this pathway compromises the lamellocyte formation, developmental delay, and host survival, suggesting its physiological relevance.Strengths:Snapshot quantification of the metabolite pool does not provide evidence that the metabolic pathway is active or not. The authors use an ex vivo isotope labelling to precisely monitor the SAM and adenosine metabolism. During infection, the methionine metabolism and adenosine recycling are upregulated, which is necessary to support the immune reaction. By combining the genetic experiment, they successfully show that the pathway is activated in immune cells.Weaknesses:The authors knocked down Ahcy to prove the importance of SAM methylation pathway. However, Ahcy-RNAi produces a massive accumulation of SAH, in addition to blocking adenosine production. To further validate the phenotypic causality, it is necessary to manipulate other enzymes in the pathway, such as Sam-S, Cbs, SamDC, etc.

We are aware of this weakness and have addressed it in a much more detailed discussion of the limitations of our study in the 6th paragraph of the Discussion.

The authors do not demonstrate how infection stimulates the metabolic pathway given the gene expression of metabolic enzymes is not upregulated by infection stimulus.

Although the goal of this work was to test by 13C tracing whether the SAM pathway activity is upregulated, not to analyze how its activity is regulated, we certainly agree with the reviewer that an explanation of possible regulation, especially in the context of the enzyme expressions we show, should be included in our work. Therefore, we have supplemented the data with methyltransferase expressions (Figure 2-figure supplement 3. And S3_Data) and better describe the changes in expression of some SAM pathway genes, which also support stimulation of this pathway by changes in expression. The enzymes of the SAM transmethylation pathway are highly expressed in hemocytes, and it is known that the activity of this pathway is primarily regulated by (1) increased methionine supply to the cell and (2) the actual utilization of SAM by methyltransferases. Therefore, a possible increase in SAM transmethylation pathway in our work can be suggested (1) by increased expression of 4 transporters capable of transporting methionine, (2) by decreased expression of AhcyL2 (dominant-negative regulator of Ahcy) and (3) by increased expression of 43 out of 200 methyltransferases. This was now added to the first section of Results.

**Recommendations for the authors:**

**Reviewing Editor Comments:**
In the discussion with the reviewers, two points were underlined as very important:(1) Knocking down Ahyc and other enzymes in the SAM methylation pathway may give very distinct phenotypes. Generalising the importance of "SAM methyaltion" only by Ahcy-RNAi is a bit cautious. The authors should be aware of this issue and probably mention it in the Discussion part.

We are aware of this weakness and have addressed it in a much more detailed discussion of the limitations of our study in the 6th paragraph of the Discussion.

(2) Sample sizes should be indicated in the Figure Legends. Replicate numbers on the RNAseq are important - were these expression levels/changes seen more than once?

Sample sizes are shown as scatter plots with individual values wherever possible and all graphs are supplemented with S1_Data table with raw data. The RNAseq data have already been described in detail in our previous paper (doi.org/10.1371/journal.pbio.3002299), but we agree with the reviewers that we should describe the necessary details again here. The replicate numbers for RNAseq data were added to figure legends, the TPM values for the selected genes shown in figures are in S1_Data and new S4_Data file with complete RNAseq data (TPM and DESeq2) was added to this revised version.

**Reviewer #1 (Recommendations for the authors):**
Major points:(1) Please provide sample sizes in the legends rather than in a supplementary table.

Sample sizes are shown either as scatter plots with individual values or added to figure legends now.

(2) More details in the methods section are needed:For hemocyte counting, are sessile and circulating hemocytes measured?

We counted circulating hemocytes (upon infection, most sessile hemocytes are released into the circulation). While for metabolomics all hemocyte types were included, for hemocyte counting we were mainly interested in lamellocytes. Therefore, we counted them 20 hours after infection, when most of the lamellocytes from the first wave are fully differentiated but still mostly in circulation, as they are just starting to adhere to the wasp egg. This was added to the Methods section.

How were levels of methionine and adenosine used in ex vivo cultures selected? This is alluded to in lines 158-159, but no references are provided.

The concentrations are based on measurements of actual hemolymph concentrations in wild-type larvae in the case of methionine, and in the case of adenosine, we used a slightly higher concentration than measured in the *adgf-a* mutant to have a sufficiently high concentration to allow adenosine to flow into the hemocytes. This is now added to the Methods section.

Minor points:

Response to all minor points: Thank you, errors has now been fixed.

(1) Line 186 - spell out MTA - 5-methylthioadenosine.(2) Lines 196-212 (and elsewhere) - spelling out cystathione rather than using the abbreviation CTH is recommended because the gene cystathione gamma-lyase (Cth) is also discussed in this paragraph. Using the full name of the metabolite will reduce confusion.

We rather used cystathionine γ-lyase as a full name since it is used only three times while CTH many more times, including figures.

(3) Figure 2 - supplement 2: please include scale bars.(4) Line 303 - spelling error: "trabsmethylation" should be "transmethylation".(5) Line 373 - spelling error: "higer" should be "higher".
**Reviewer #2 (Recommendations for the authors):**
For the RNAseq data, it's unclear whether the gene expression data in Figures 2 and 4 include biological replicates, so it's unclear how much weight we should place on them.

The replicate numbers for RNAseq data were added to figure legends, the TPM values for the selected genes shown in figures are in S1_Data and new S4_Data file with complete RNAseq data (TPM and DESeq2) was added to this revised version.

The representation of these data is also a weakness: Figure 2 shows measurements of transcripts per million, but we don't know what would be high or low expression on this scale.

We have added the actual TPM values for each cell in the RNAseq heatmaps in Figure 2, Figure 2-figure supplement 3, and Figure 4 to make them more readable. Although it is debatable what is high or low expression, to at least have something for comparison, we have added the following information to the figure legends that only 20% of the genes in the presented RNAseq data show expression higher than 15 TPM.

Figure 4 is intended to show expression changes with treatment, but expression changes should be shown on a log scale (so that increases and decreases in expression are shown symmetrically) and should be normalized to some standard level (such as uninfected lamellocytes).

The bars in Figure 4C,D show the fold change (this is now stated in the y-axis legend) compared to 0 h (=uninfected) Adk3 samples - the reason for this visualization is that we wanted to show (1) the differences in levels between Adk3 and Adk2 and in levels between Ak1 and Ak2, respectively, and at the same time (2) the differences between uninfected and infected Adk3 and Ak1. In our opinion, these fold change differences are also much more visible in normal rather than log scale.

**Reviewer #3 (Recommendations for the authors):**
(1) It might be interesting to test how general this finding would be. How about Bacterial or fungal infection? The authors may also try genetic activation of immune pathways, e.g. Toll, Imd, JAK/STAT.

Although we would also like to support our results in different systems, we believe that our results are already strong enough to propose the final hypothesis and publish it as soon as possible so that it can be tested by other researchers in different systems and contexts than the *Drosophila* immune response.

(2) How does the metabolic pathway get activated? Enzyme activity? Transporters? Please test or at least discuss the possible mechanism.

The response is already provided above in the Reviewer #3 (Public review) section.

(3) The authors might test overexpression or genetic activation of the SAM transmethylation pathway.

Although we agree that this would potentially strengthen our study, it may not be easy to increase the activity of the SAM transmethylation pathway - simply overexpressing the enzymes may not be enough, the regulation is primarily through the utilization of SAM by methyltransferases and there are hundreds of them and they affect numerous processes.

(4) Supplementation of adenosine to the Ahcy-RNAi larvae would also support their conclusion.

Again, this is not an easy experiment, dietary supplementation would not work, direct injection of adenosine into the hemolymph would not last long enough, adenosine would be quickly removed.

(5) It is interesting to test genetically the requirement of some transporters, especially for gb, which is upregulated upon infection.

Although this would be an interesting experiment, it is beyond the scope of this study; we did not aim to study the role of the SAM transmethylation pathway itself or its regulation, only its overall activity and its role in adenosine production.